# THE PERSIAN RUG: SOLVING TOY MODELS OF SUPERPOSITION USING LARGE-SCALE SYMMETRIES

## ABSTRACT

We present a complete mechanistic description of the algorithm learned by a minimal non-linear sparse data autoencoder in the limit of large input dimension. The model, originally presented in Elhage et al. (2022), compresses sparse data vectors through a linear layer and decompresses using another linear layer followed by a ReLU activation. We notice that when the data is permutation symmetric (no input feature is privileged) large models reliably learn an algorithm that is sensitive to individual weights only through their large-scale statistics. For these models, the loss function becomes analytically tractable. Using this understanding, we give the explicit scalings of the loss at high sparsity, and show that the model is near-optimal among recently proposed architectures. In particular, changing or adding to the activation function any elementwise or filtering operation can at best improve the model's performance by a constant factor. Finally, we forward-engineer a model with the requisite symmetries and show that its loss precisely matches that of the trained models. Unlike the trained model weights, the low randomness in the artificial weights results in miraculous fractal structures resembling a Persian rug, to which the algorithm is oblivious. Our work contributes to neural network interpretability by introducing techniques for understanding the structure of autoencoders.

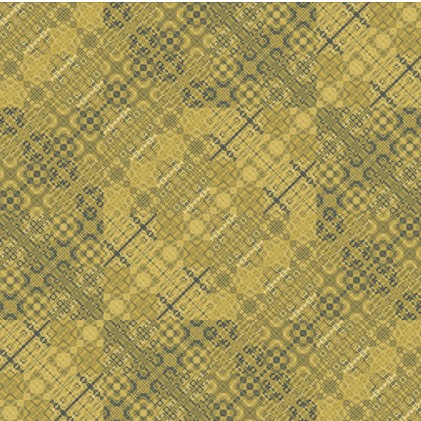

Figure 1: The Persian Rug, an artificial set of weights matching trained model performance.

## 1 INTRODUCTION

Large language model capabilities and applications have recently proliferated. As these systems advance and are given more control over basic societal functions, it becomes imperative to ensure their reliability with absolute certainty. Mechanistic interpretability aims to achieve this by obtaining a concrete weight-level understanding of the algorithms learned and employed by these models. A major impediment to this program has been the difficulty of interpreting intermediate activations. This is due to the phenomena of superposition, in which a model takes advantage of sparsity in the input data to reuse the same neurons for multiple distinct features, obscuring their function. Finding

a systematic method to undo superposition and extract the fundamental features encoded by the network is a large and ongoing area of research Bricken et al. (2023); Cunningham et al. (2023); Gao et al. (2024); Engels et al. (2024).

Currently, the most popular method of dealing with superposition is dictionary learning with sparse autoencoders. In this method, the smaller space of neuron activations at a layer of interest is mapped to a larger feature space. The map is trained to encourage sparsity and often consists of an affine + ReLU network. This method has been applied to large language models revealing many strikingly interpretable features (e.g. corresponding to specific bugs in code, the golden gate bridge, and sycophancy), even allowing for a causal understanding of the model's reasoning in certain scenarios Marks et al. (2024).

The sparse decoding ability of the affine + ReLU map was recently studied in the foundational work Elhage et al. (2022), which introduced and studied a toy model of superposition. The model consisted of a compressing linear layer modeling the superposition[1] followed by a decompressing affine + ReLU layer, trained together to auto-encode sparse data. They showed that the network performs superposition by encoding individual feature vectors into nearly orthogonal vectors in the smaller space. The affine layer alone is unable to decode sparse input vectors sufficiently well to make use of superposition, but the addition of the ReLU makes it possible by screening out negative interference.

While Elhage et al. (2022) provides valuable empirical and theoretical insights into superposition, it does not obtain a strong enough description of the model algorithm to quantitatively characterize the algorithm's performance. Given the extensive use of the affine + ReLU map for decoding sparse data in practice, it is important to obtain a complete analytic understanding of the model behavior over a large parameter regime. As we will see, this will inform the design of better sparse autoencoder architectures.

In this work we obtain such an understanding by considering a particularly tractable regime of the Elhage et al. (2022) model: permutation symmetric data (no input feature is privileged in any way), and the thermodynamic limit (a large number of input features), while maintaining the full range of sparsity and compression ratio values. In this regime, the learned model weights are permutation symmetric on large scales, which sufficiently simplifies the form of the loss function to the point where it is analytically tractable, leaving only a small number of free parameters. We then forwards-engineer an artificial set of weights satisfying these symmetries and optimizing the remaining parameters, which achieves the same loss as a corresponding trained model, implying that trained models also implement the optimal permutation symmetric algorithm. The artificial set of weights resembles a Persian rug fig. 1, whose structure is a relic of the minimal randomness used in the construction, illustrating that the algorithm relies entirely on large-scale statistics that are insensitive to this structure. Finally, we derive the exact power-law scaling of the loss in the high-sparsity regime.

We expect our work to impact the field of neural network interpretability in multiple ways. First, our work provides a basic theoretical framework that we believe can be extended to other regimes of interest, such as structured correlations in input features, which may help predict scaling laws in the loss based on the data's correlations. Second, our work rules out a large class of performance improvement proposals for sparse autoencoders. Finally, our work provides an explicit example of a learned algorithm that is insensitive to microscopic structure in weights, which may be useful for knowing when not to analyze individual weights.

The paper is structured as follows. In section 2 we review the model and explain our training procedure. In section 3 we show empirically that large models display a "statistical permutation symmetry". In section 4 we extract the algorithm by plugging the symmetry back into the loss, introduce the Persian Rug model which optimizes the remaining parameters, show that large trained models achieve the same loss, and derive the loss behavior in the high sparsity limit. In section 5 we conclude and discuss related works.

---

[1]This is because, if good enough recovery is possible for most features, the pigeonhole principle tells us that at least some of the smaller space activations must encode information about multiple input features.

## 2 THE MODEL

We study the following non-linear autoencoder with parameters $W_{\mathrm{in}} \in \mathbb{R}^{n_d \times n_s}, W_{\mathrm{out}} \in \mathbb{R}^{n_s \times n_d}, \mathbf{b} \in \mathbb{R}^{n_s}$ with $n_d \leq n_s$,

$$f_{\mathrm{nonlinear}}(\mathbf{x}) = \mathrm{ReLU}(W_{\mathrm{out}} W_{\mathrm{in}} \mathbf{x} + \mathbf{b}). \tag{1}$$

Here, $W_{\mathrm{in}}$ is an encoding matrix which converts a sparse activation vector $\mathbf{x}$ to a dense vector, while $W_{\mathrm{out}}$ perform the linear step of decoding. We also consider a simple model for the sparse data on which this autoencoder operates. We work with data that is permutation symmetric in the sense that $(x_1, ..., x_n)$ is equal in distribution to $(x_{\pi(1)}, x_{\pi(2)}, ..., x_{\pi(n)})$ for any permutation $\pi$. Each vector $\mathbf{x}$ is drawn i.i.d. during training, and each component is drawn i.i.d. with $x_i = c_i u_i$, where $c_i \sim \mathrm{Bernoulli}(p)$ and $u_i \sim \mathrm{Uniform}[0, 1]$ are independent variables. This ensures that $\mathbf{x}$ is sparse with typically only $p n_s$ features turned on.

We train our toy models to minimize the expected $L_2$ reconstruction loss,

$$L(\mathbf{x}; W_{\mathrm{out}}, W_{\mathrm{in}}, \mathbf{b}) = n_s^{-1} \mathbb{E} ||\mathbf{x} - f_{\mathrm{nonlinear}}(\mathbf{x})||_2^2. \tag{2}$$

It is known that for the linear model (eq. (1) without the ReLU), the optimal solution is closely related to principle component analysis (see, for example, Plaut (2018) and p. 563 or Bishop & Nasrabadi (2006)). In particular, the reconstruction loss decreases linearly in the hidden dimension $n_d$ when all features are i.i.d. On the other hand, the model eq. (1) will have a much quicker reduction in loss, as will be described in section 3.1.

We train all models with a batch size of 1024 and the Adam optimizer to completion. That is training continues as long as the average loss over the past 200 batches is lower than the average loss over the 200 batches prior to that one. Our goal with training is to ensure that we have found an optimal model in the large-data limit to analyze the structure of the model itself. See also appendix G for more training details.

## 3 EMPIRICAL OBSERVATIONS

In this section, we present empirical observations of the trained models. We start by presenting a remarkable phenomenon this model exhibits in the high-sparsity regime: a dramatic decrease in loss as a function of the compression ratio. We then turn to a mechanistic interpretation of the weights which gives empirical evidence for the phenomena needed to understand the algorithm the model learns. These are manifestations of a partially preserved permutation symmetry of the sparse degrees of freedom.

### 3.1 FAST LOSS DROP

To gauge the performance of the linear and non-linear models, we plot the loss (eq. (2)) as a function of the compression ratio $n_d/n_s$. In fig. 2, we see that the non-linear model dramatically outperforms the linear model and that the out-performance begins already for small ratios and continues up until much larger ratios (they must coincide again in the trivial case $n_d/n_s = 1$). In this paper, we will see that the slope and duration of this initial fall is controlled by $p$ (section 4.3). In particular, in the high-sparsity regime ($p$ close to zero), the loss drops very quickly near the $n_d/n_s \approx 0$ regime. To explain this behavior, we analyze the algorithm the model encodes.

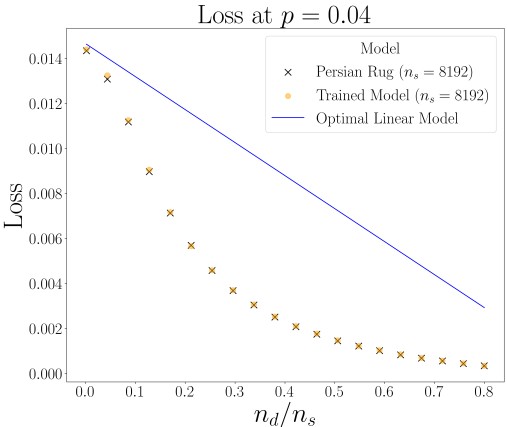

Figure 2: Loss curves of trained models, Persian Rug models, and optimal linear models as a function of the compression ratio.

## 3.2 STATISTICAL PERMUTATION SYMMETRY

Rather than looking individually at the weights, it is helpful to look at the matrix $W = W_{\text{out}}W_{\text{in}}$, shown in fig. 3. When we rewrite eq. (1) in terms of $W$,

$$(f_{\text{nonlinear}}(\mathbf{x}))_i = \text{ReLU}(W_{ii}x_i + \sum_{j=1, j\neq i}^{n_s} W_{ij}x_j + \mathbf{b}_i),$$

we see that the quantity $W_{ij}$ measures how much feature $i$'s reconstruction "listens" to feature $j$. Given the data has a permutation symmetry[2], so that there is no reason for the reconstruction of $x_i$ to listen any more to $x_j$ than to $x_k$ for $i \neq j \neq k$, we might expect a similar symmetry to manifest in $W$. For example, we might expect $W_{ij} = W_{ik} = W_{ji}$ and $W_{ii} = W_{jj}$ for $i \neq j \neq k$. Figure 3 shows this is not the case, but in this section we show empirically that $W$ will satisfy a weaker "statistical permutation symmetry" when $n_s$ gets large. More precisely, in the large $n_s$ regime, $W$ will satisfy the following properties:

a) the diagonal elements become the same (fig. 4),

b) the bias elements become the same and uniformly negative, which can be seen in the uniformity and slight blue shade in fig. 3 and is quantified in fig. 5,

c) the off-diagonal terms are sufficiently uniform to motivate a Gaussian approximation to a reconstruction error term $\nu_i := W_{ii}^{-1} \sum_{j\neq i} W_{ij}x_j$ defined for each row (fig. 8), and finally,

d) the corresponding Gaussians are equal in distribution across rows (fig. 6 and fig. 7).

We confirm that each of the properties listed in the definition of statistical permutation symmetry above hold empirically in figs. 4, 5, 7 and 8. Each of these figures contains three subfigures (corresponding to $n_s \in \{128, 1024, 6182\}$), to show that the relevant property manifests as $n_s$ gets large. For example, in fig. 4, we show that the root mean square variation of the diagonals of $W$,

$$\Delta\text{diag}(\boldsymbol{W}) := n_s^{-1} \sum_{i=1}^{n_s} (W_{ii} - \overline{\text{diag}(W)})^2$$

(where $\overline{\text{diag}(W)} = n_s^{-1} \sum_{i=1}^{n_s} W_{ii}$) tends to zero for various $r$ and $p$. (We refer to this as the root mean square deviation instead of the standard deviation to emphasize the fact that $W$ is not considered to be a random variable in our analysis. Throughout this paper, we reserve terms like mean and standard deviation for random variables.) Similar quantities and plots for items b) and d) can be found in appendix A.

[2]By this we mean $(x_1, ..., x_n)$ is equal in distribution to $(x_{\pi(1)}, ..., x_{\pi(n)})$ for any permutation $\pi$

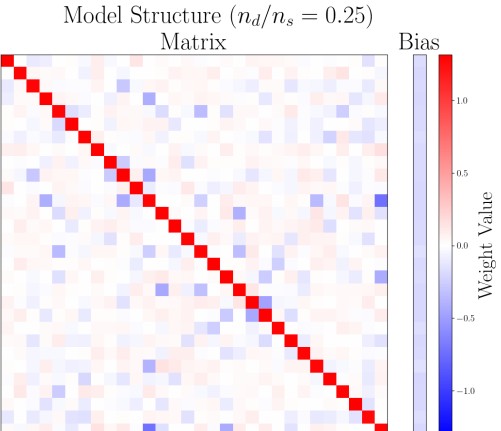

Figure 3: Plot of the first $30 \times 30$ $W$ elements and the corresponding bias (**b**) components, at $p = 4.5\%$ and ratio 0.25 ($n_s = 512$). The diagonal components are all at similar values of $1.29 \pm .01$ (one standard deviation) while the off-diagonal components are approximately mean-zero, appearing like noise. The bias elements are all negative around $-.18 \pm .01$. This statistical uniformity is a permutation symmetry across the sparse features.

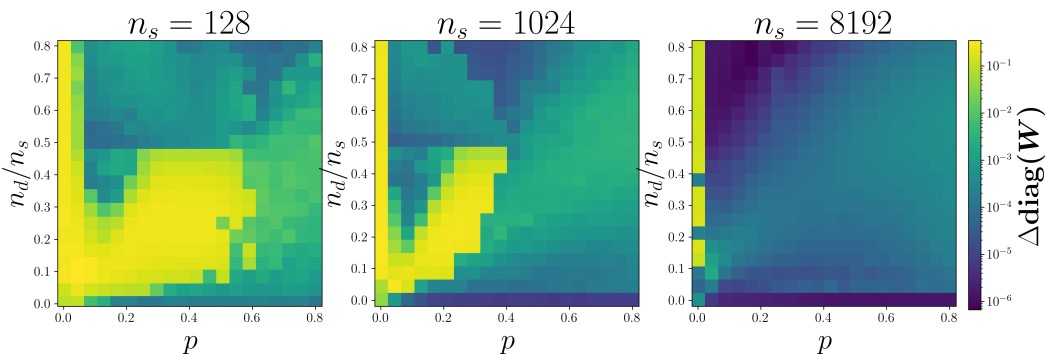

Figure 4: Permutation symmetry of diagonal values. We plot the mean-square fluctuation of the diagonal values corresponding to each model. Models are trained as a function of $p$ and $n_d/n_s$. The emergence of symmetry as $n_s$ grows (at all locations in the diagram) is a crucial element of the algorithm implemented by the autoencoders.

We now discuss the Gaussianity property (item c)) in more detail. For a fixed (deterministic) $W$ and random $x$, the quantity $\nu_i$ is a random variable which measures the extent to which the pre-activation reconstruction of $x_i$ erroneously receives signal from other components $x_j$ with $j \neq i$. Given that $\nu_i$ is a sum of $n_s$ independent random variables, it is natural to ask how well it can be approximated by a Gaussian in the large $n_s$ regime. Clearly some conditions on $W_{ij}$ and $x_j$ will be needed. The Berry-Esseen theorem for independent random variables implies that, if the $x_j$ have finite third moment, the quantity

$$\Lambda := \max_i \frac{\sum_{j \neq i} |W_{ij}|^3}{\left(\sum_{j \neq i} W_{ij}^2\right)^{3/2}}, \tag{3}$$

up to a constant factor that depends on the first three moments of $x$, is a measure of how far $\nu_i$ is from Gaussianity (see, for example, Petrov (1972)). In fig. 8, we plot $\Lambda$ and show it tends to 0 as $n_s$ grows larger. We leave as an open theory problem the identification of conditions that would guarantee a central limit theorem for a sequence $\nu_i^{(k)}$ defined by $W^{(k)}$ that optimize eq. (1) on a growing sequence of problems (e.g. as $n_s \to \infty$ with $n_d/n_s = r$).

## 3.3 Optimization of residual parameters

The statistical permutation symmetry places constraints on the possible values of $W$ and $\mathbf{b}$. The constraint on $\mathbf{b}$ is straightforward: it is proportional to the all ones vector, i.e. there is a number $b$ such that $\mathbf{b}_i = b$ for all $i$. As will be explained in section 4.1, the relevant degrees of freedom remaining in $W$ are a number $a$ equal to the diagonals ($W_{ii} = a$ for all $i$) and another number $\sigma$ characterizing the variance of $\nu_i$ (var $\nu_i = \sigma$ for all $i$). The precise values of the off diagonals can be thought of as irrelevant "microscopic information". Thus there are three relevant degrees of freedom remaining: $b, a$, and $\sigma$.

In section 4.2.2, we give a specific set of values for $W_{\text{in}}$ and $W_{\text{out}}$ via the "Persian Rug" matrix, which have the statistical permutation symmetry in the $n_s \to \infty$ limit while also optimizing $\sigma$. The remaining parameters, $a$ and $b$ can be optimized numerically. In fig. 2 we compare the loss curve of this artificial model with that of a trained model, and see that they are essentially the same.

## 4 Extracting the Algorithm

In this section, we give a precise explanation of the algorithm the model performs. We start with a qualitative description of why the statistical permutation symmetry gives a good auto-encoding algorithm when the remaining macroscopic degrees of freedom are optimized. We then find an artificial set of symmetric weights with optimized macroscopic parameters. We show that the trained models achieve the same performance as the artificial model, thus showing they are optimal even restricting to statistically symmetric solutions. Finally, we derive an explicit form of the loss scaling and argue that ReLU performs near optimally among element-wise or "selection" decoders.

## 4.1 Qualitative Description

A key simplification is to consider strategies as collections of low-rank affine maps rather than as the collection of weights directly. In other words, consider the tuple $(W, \mathbf{b})$ where $W = W_{\text{out}}W_{\text{in}}$ to define the strategy. We must restrict to $W$ with rank no more than $n_d$ because it is the product of two low-rank matrices. Given any such $W$ we may also find $W_{\text{in}}$ and $W_{\text{out}}$ of the appropriate shape (e.g. by finding the SVD), so the two representations are equivalent.

We now write the output for feature $i$ in terms of $W$ under the statistical permutation symmetry assumption motivated in section 3. We have

$$(f_{\text{nonlinear}}(\mathbf{x}))_i = \text{ReLU}(W_{ii}x_i + \sum_{j=1, j \neq i}^{n_s} W_{ij}x_j + \mathbf{b}_i) = \text{ReLU}\left(a(x_i + \nu_i) + b\right) \quad (4)$$

where we have used our assumptions that $W_{ii} = a$ and $b_i = b$. We also assume that the $\nu_i$ are Gaussian and are all equal in distribution. From this, we have

$$(f_{\text{nonlinear}}(\mathbf{x}))_i \overset{\mathcal{D}}{=} \text{ReLU}\left(a(x_i + \nu) + b\right),$$

where $\overset{\mathcal{D}}{=}$ denotes equality in distribution and $\nu \sim N(0, \sigma^2)$ (we may assume $\nu$ is mean zero because its mean can be absorbed into the bias $b$).

The expected reconstruction error is therefore

$$L = \mathbb{E}_{x,\nu}[(x - \text{ReLU}\left(a(x + \nu) + b\right))^2] \quad (5)$$

where $x = cu$ with $c \sim \text{Bernoulli}(p)$ and $u \sim \text{Uniform}[0, 1]$. We can further decompose this by taking the expectation value over whether $x$ is "on" or "off" ($c = 1$ or $c = 0$, respectively). That is,

$$L = (1 - p)L_{\text{off}} + pL_{\text{on}},$$

where

$$L_{\text{off}} = \mathbb{E}_\nu[(\text{ReLU}\left(a\nu + b\right))^2], \text{ and}$$

$$L_{\text{on}} = \mathbb{E}_{u,\nu}[(u - \text{ReLU}\left(a(u + \nu) + b\right))^2].$$

Let us qualitatively explore the regimes of macroscopic parameters $a, b, \sigma$ when either or both of

these loss terms are low.

The impact of all the non-diagonal terms has been summed up in the "noise" $\nu$. Though it is a deterministic function of $\mathbf{x}$, output $i$ has no way to remove $\nu$ from $u + \nu$. The best it can do is to estimate $u$ from $u + \nu$ because it doesn't have any other information. This exemplifies a key principle – by restricting the computational capacity of our decoder, deterministic, but complicated correlations act like noise.

The main advantage of the nonlinear autoencoder is that the dominant contribution to the loss, $L_{\text{off}}$, can be immediately screened away by making $a\nu + b$ either small or negative, allowing the network to focus on encoding active signals. This immediate screening is always possible either by choosing $b$ large and negative or $a$ and $b$ small. However, these strategies come at a cost because the output value is distorted from $u$ to $au + b$. It is thus preferable instead for $\nu$ and $b$ to be as small as possible, which occurs when $\sigma$ is as small as possible. As we will see, $\sigma$ is the only parameter we are not free to choose in the large $n_s$ limit, and whose value will be bounded as a function of $n_d/n_s$ and $p$. Since $L_{\text{off}}$ is the dominant contribution to the loss, therefore, it will thus be necessary to damp the signal by setting $a$ small and/or $b$ large and negative in regimes where $\sigma$ is uncontrollably large.

Given that we see a statistical permutation symmetry in trained models let's consider symmetric strategies so that $W_{ii} = a$ and $\mathbf{b}_i = b$ for all $i = 1, \ldots, n_s$. We will show that optimizing the remaining macroscopic parameters makes $f_{\text{nonlinear}}$ act close to the identity on sparse inputs.

## 4.2 OPTIMIZING THE MACROSCOPIC PARAMETERS

We have seen qualitatively that a statistically symmetric strategy exists in certain regimes of the macroscopic parameters. Two of these parameters, $a$ and $b$ are unconstrained. Furthermore the loss should be monotonically increasing with $\sigma$ because a larger $\sigma$ implies more noise which hinders reconstruction. Thus we now prove lower bounds on $\sigma$ and construct an artificial set of statistically permutation symmetric weights which achieve this bound. Finally we will compare the reconstruction loss of this strategy with the learned one to justify that those ignored microscopic degrees of freedom were indeed irrelevant.

### 4.2.1 OPTIMAL $\sigma$

Assuming the permutation symmetry we discovered earlier in our empirical investigations we will derive a bound on the variance of the output. Additionally for an optimal choice of $\mathbf{b}$ the average loss is increasing in the variance, because a larger variance corresponds to a smaller signal-to-noise ratio. Taken together these two facts will give a lower bound on the loss. We will then provide an explicit construction which achieves this lower bound and illustrates how the algorithm works.

The lower bound on the variance comes from the fact that $W$ is low-rank with constant diagonals. For now let us ignore the overall scale of $W$, and just rescale so that the diagonals are exactly 1. The bound we are about to prove is very similar to the Welch bound (Welch, 1974) who phrased it instead as a bound on the correlations between sets of vectors. We produce an argument for our context, which deals with potentially non-symmetric matrices $W$, the details of which are located in appendix D.

We show that

$$\sigma^2 \geq \frac{4p - 3p^2}{12} \left( \frac{n_s}{n_d} - 1 \right) \tag{6}$$

with equality only when $W$ is symmetric, maximum rank, with all non-zero eigenvalues equal. This naturally leads to a candidate for the optimal choice of $W$, namely matrices of the form

$$W \propto OPO^T \text{ and } W_{ii} = 1 \tag{7}$$

where $O$ is an orthogonal matrix and $P$ is any rank-$n_d$ projection matrix. This kind of matrix saturates the bound because it is symmetric and has all nonzero eigenvalues equal to 1.

A note on the connections between these matrices and tight frames – if we take the "square root" of $W$ as a $n_s \times n_d$ matrix $\sqrt{W}$ such that $\sqrt{W}\sqrt{W}^\dagger = W$ then the $n_d$ dimensional rows of $\sqrt{W}$ are a tight-frame. This is because $\text{Tr}(W)^2 = n_s^2 = n_d \, \text{Tr} \, WW^\dagger$ which is the variational characterization of tight-frames as in Theorem 6.1 from Waldron (2018).

### 4.2.2 PERSIAN RUG MODEL

We now give an explicit construction of an optimal choice for $W$. The construction is based on a Hadamard matrix of size $n_s$. A square matrix $H$ is a Hadamard matrix if $H_{ij} \in \{-1, 1\}$ and its rows are orthogonal (see, for example, Horadam (2012)).

We then define a Persian Rug matrix as

$$R_{ij} = n_d^{-1} \sum_{k \in S} H_{ik} H_{jk}$$

where $S \subset \{1, ..., n_s\}$ with $|S| = n_d$. In fig. 1 we plot such a matrix for $n_s = 256$, $n_d = 40$, and $S$ chosen randomly. The matrix $R$ has diagonals equal to 1 because each diagonal is the average of $n_d$ terms of the form $(H_{ik})^2 = (\pm 1)^2 = 1$. The matrix $R$ is a projector because the rows of $H$ are orthogonal, and therefore $R$ is the sum of commuting rank-1 projectors. Therefore $R$ saturates eq. (6). Furthermore, one can readily check that it exactly satisfies the symmetry for off-diagonal terms as well as shown in appendix F, which we direct readers to for a further discussion of $R$. There remain two variables to optimize, $a$ and $b$ (recall $\mathbb{E}[\nu]$ can be absorbed into $b$). We do this numerically and compare to a trained model in fig. 2 (details on the training process can be found in appendix G).

### 4.3 LOSS SCALING AT HIGH SPARSITY

Having obtained a simple expression for the loss in terms of constants $a, b$ and two simple random variables $x \sim \text{Uniform}[0, 1]$ and $\nu \sim \mathcal{N}(0, \sigma)$, as well has having deduced an achievable lower bound for $\sigma$, we are now able to explain why the simple ReLU model performs so well at high sparsity. For ease of notation let us use $r = n_d/n_s$.

### 4.3.1 INITIAL LOSS (RATIO=0)

Let us first consider the $r \to 0$ limit with all other parameters fixed. Then $\sigma \to \infty$ because of the bound in eq. (6) so the fluctuations in $\nu$ overwhelms the signal term. This means that the optimal $a$ is

$$a = p \frac{\mathbb{E}_{u,\nu}\left[u\,\text{ReLU}(\nu + b)\right]}{\mathbb{E}_\nu\left[\text{ReLU}(\nu + b)^2\right]} + O(\sigma^{-1}). \tag{8}$$

The loss then becomes

$$L = (1 - p)a^2 \mathbb{E}_\nu[\text{ReLU}\left(\nu + b\right)^2] + p\mathbb{E}_{u,\nu}[(u - a\,\text{ReLU}\left(\nu + b\right))^2] + O(\sigma^{-1})$$

$$= a^2 \mathbb{E}_\nu[\text{ReLU}\left(\nu + b\right)^2] - 2ap\mathbb{E}_{u,\nu}[u\,\text{ReLU}(\nu + b)] + p\mathbb{E}_u[u^2] + O(\sigma^{-1})$$

plugging in $a$ explicitly gives

$$L = p\mathbb{E}_u[u^2] - p^2 \frac{\left(\mathbb{E}_u\left[u\right]\right)^2 \left(\mathbb{E}_\nu\left[\text{ReLU}(\nu + b)\right]\right)^2}{\mathbb{E}_\nu\left[\text{ReLU}(\nu + b)^2\right]} + O(\sigma^{-1}).$$

Thus we can conclude that

$$\lim_{r \to 0} L = p\mathbb{E}_u[u^2] + O(p^2) = \frac{p}{3} + O(p^2)$$

Thus we see that in the $p \ll 1$ regime we have $L \to L_0(p) \sim O(p)$ independent of the other parameters. We will now see that increasing $r$ will quickly cause the loss to drop to $O(p^2)$.

### 4.3.2 DERIVING THE LOSS SCALING

In appendix E.1 we derive an upper bound on the loss function by plugging in appropriate ansatze for $a$ and $b$. We find that

$$L < O\left(\sigma^2 p \log \frac{1}{p}\right) \sim O\left(\frac{p^2}{r} \log \frac{1}{p}\right), \tag{9}$$

when $p$ is small and when $r \gg p$.

On the other hand, in appendix E.2 we also derive a lower bound in the high-sparsity limit $L > O(p^2/r)$ in the high sparsity limit up to logarithmic corrections. We show this in fact holds for a more general class of activation functions. In particular, any function which acts element-wise or filters out elements will give an on-loss contribution of the form

$$\mathbb{E}_{u,\nu}[(u - f(u + \nu + b))^2]$$

which has a lower bound due to $\nu$ destroying information about $u$. Thus we can conclude that

$$L \sim O\left(\frac{p^2}{r}\right)$$

up to logarithmic factors whenever $p/r \ll 1$. It is sensible that the loss function scales inversely with compression ratio.

## 5 RELATIONSHIP TO OTHER WORKS

### 5.1 AUTOENCODERS

Our work focuses specifically on sparse autoencoders, and encoding sparse data, which is parallel to work explaining the dynamics and emergence of feature learning in autoencoders. Refinetti & Goldt (2023) show that shallow autoencoders learn the principal components of the data sequentially and Nguyen (2021) shows a similar dynamical result via a mean-field analysis. It's seen that such autoencoders function even in a regime where the number of features and the size of the input are proportional with numerical evidence for Gaussian universality (Shevchenko et al., 2023). This universality is shown for shallow in auto-encoders following gradient dynamics (Kögler et al., 2024).

#### 5.1.1 MECHANISTIC INTERPRETABILITY AND SPARSE AUTOENCODERS

Mechanistic interpretability is a research agenda which aims to understand learned model algorithms through studying their weights (see Olah et al. (2020) for an introduction). Recent results relating to language models include Meng et al. (2023), which finds a correspondence between specific facts and feature weights, along with Olsson et al. (2022), which shows that transformers learn in context learning through the mechanism of "induction heads".

A key issue for the agenda of mechanistic interpretability is that the model stores features in superposition. Elhage et al. (2022) introduced the toy model of superposition we study in this paper. While that work focused on mapping empirically behaviors of the model in multiple regimes of interest such as correlated inputs, we focused on a regime with enough symmetry to solve the model analytically given observed symmetries in trained models. Chen et al. (2023) study this model in the context of singular learning theory. As part of their work, they characterize the loss using a different high sparsity approximation than the one we present in this paper (they assume exactly one input feature is on). Then they derive a subset of the critical points and their corresponding local learning coefficients under the assumption $n_d = 2$. Refinetti & Goldt (2023) study the learning dynamics of the same model but without the sparsity assumption.

One way to extract interpretable features that are stored in superposition is through dictionary learning. While the concept of dictionary learning was introduced by (Mallat & Zhang, 1993), the practical use of sparse autoencoders to understand large language models has accelerated recently due to mezzo-scale open weight models (Gao et al., 2024; Lieberum et al., 2024) and large-scale open-output models Bricken et al. (2023). These features are highly interpretable (Cunningham et al., 2023) and scale predictably. Interestingly, the scaling is quite similar for the various different architectures they consider, differing primarily by a constant, which fits with the predictions in this work.

Our study of Elhage et al. (2022)'s model of superposition lend some insight into the dictionary learning problem. In particular, we have seen that the dominant source of error is not from determining which features are present, but rather the actual values of those features. Small modifications to the activation functions, such as gating Rajamanoharan et al. (2024), k-sparse Makhzani & Frey (2013), or TRec non-linearity Taggart (2024); Konda et al. (2014), are insufficient to fix this problem

as they do not solve the basic issue of noisy outputs. In this context our work implies that innovative architectures, that are suitable both for gradient-based training and also for decoding sparse features, must be developed.

While our work focuses on Elhage et al. (2022) toy model of superposition, one can study the dictionary learning with sparse autoencoders problem directly under various models for the data and various algorithms (see, for example, Rangamani et al. (2017); Nguyen et al. (2019); Arora et al. (2015); Agarwal et al. (2016); Spielman et al. (2012)). For these problems, we suppose we are given data vectors $y_i$ generated by $y_i = A^* x_i$, where $A^* \in \mathbb{R}^{n_d \times n_s}$ is the "true" dictionary and $x_i$ are parametrically sparse vectors. The goal is then to recover $A^*$ and the $x_i$. Under various assumptions on $A^*$ and the $x_i$, one can prove various desirable results for various algorithms for estimating them. For example, in Rangamani et al. (2017) and Nguyen et al. (2019) it is assumed that $A^*$ has unit columns and is *incoherent*, meaning that its columns $\{A_i^*\}$ have inner-products bounded by

$$\max_{i \neq j} |\langle A_i^*, A_j^* \rangle| \leq \frac{\mu}{\sqrt{n}}.$$

These authors then give convergence results for learning the model

$$\hat{y} = V^T ReLU(Vy - \epsilon) \tag{10}$$

with $\epsilon$ a learnable bias and $V \in \mathbb{R}^{n_s \times n_d}$ learnable weights. In particular, Rangamani et al. (2017) shows that the support of $x$ can be recovered for sufficiently sparsity and incoherence and that $A^*$ is critical point for $V$ in the loss landscape; Nguyen et al. (2019) shows that eq. (10) trained with gradient descent recovers the true dictionary in certain parameter regimes.

In contrast, the neural networks in our work search over the space of dictionaries to find ones that encode sparse information in a way particularly suitable for reconstruction by a single linear + ReLU layer. As a result, the dictionary our network finds contains *additional structure optimized for a particular recovery process*. For this reason, we find that the relevant error parameter is not the incoherence (in our notation) $\max_{i \neq j} |W_{ij}|$ but rather the variance of off-diagonal elements in each row $\sum_j W_{ij}^2$, and that it is this parameter that needs to be minimized for a given compression ratio.

### 5.1.2 COMPRESSED SENSING, STATISTICAL PHYSICS

It is known that compressed sparse data can be exactly reconstructed by solving a convex problem (Candes & Tao, 2005; Candes et al., 2006; Donoho & Elad, 2003; Donoho, 2006) given knowledge of the compression matrix. Furthermore, using tools from statistical physics it is possible to show that this holds for typical compressed sparse data (Ganguli & Sompolinsky, 2010). Learning the compression matrix is also easy in certain circumstances(Sakata & Kabashima, 2013). For a more general review on compressed sensing and it's history consider the introduction by Davenport et al. (2012). The reconstruction procedure typically used in compressed sensing is optimizing a (convex) relaxation of finding the sparsest set of features which reproduces your data vector. This is significantly different to the setting of sparse autoencoders which try to obtain the sparse features using only one linear + activation layer.

The discrepancy between the ability of convex optimization techniques to achieve zero loss while a linear + ReLU model necessarily incurs an error suggests that a more complex model architecture is needed for sparse autoencoders when it is desirable to calculate the feature magnitude to high precision. This may occur, for example, if one wishes to insert a sparse autoencoder into a model without corrupting its downstream outputs. An important line of work is algorithms based on message-passing schemes brought to fame by Donoho et al. (2009), and extended to more general encoding matrices by Rangan et al. (2019), a more general encoding scheme (Schniter et al., 2016), for ill conditioned matrices (Ma & Ping, 2017), and proved without statistical physics methods by Takeuchi (2019). These works may hold the key to improving interpretability, particularly for downstream tasks such as circuit recovery.

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

# A   CONVERGENCE FIGURES OF SECTION 3.2

Generalizing from Section 3.2, let $\Delta : \mathbb{R}^{n_s} \to \mathbb{R}^+$ be the root mean square operator

$$\Delta \boldsymbol{v} := n_s^{-1} \sum_{i=1}^{n_s} (v_i - \overline{\boldsymbol{v}})^2,$$

where $\overline{\boldsymbol{v}} = n_s^{-1} \sum_{i=1}^{n_s} \boldsymbol{v}_i$. With this definition, let $\boldsymbol{\nu}$ be the vector with $\boldsymbol{\nu}_i = \nu_i$.

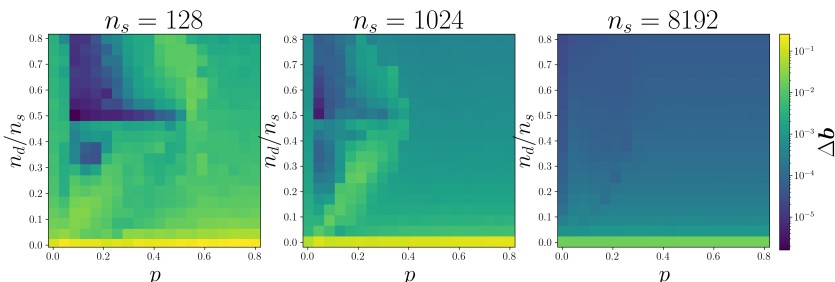

Figure 5: Permutation symmetry of bias values. We plot the mean-square fluctuation of values in the bias vectors corresponding to each model, which are trained as a function of $p$ and $n_d/n_s$. As $n_s$ increases the fluctuation over bias elements generally decreases in all trained models.

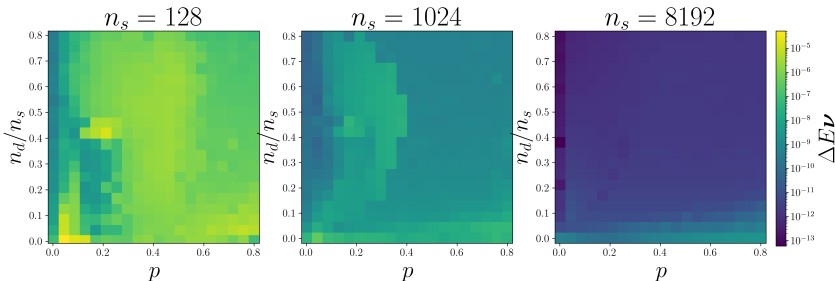

Figure 6: Permutation symmetry of $E\boldsymbol{\nu}$. In this case, $E\boldsymbol{\nu}$ is given by the off-diagonal row-sums, scaled by $p$.

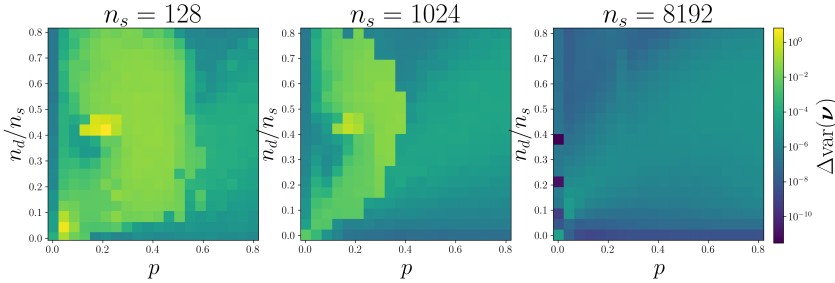

Figure 7: Permutation symmetry of var($\boldsymbol{\nu}$). The symmetry breaking parameter $\Delta \operatorname{var}(\boldsymbol{\nu})$ is given by the variance across all rows of the squared sum of the off diagonal elements in each row, up to a constant. Once $n_s$ reaches 8192 all noise variables have nearly identical variances.

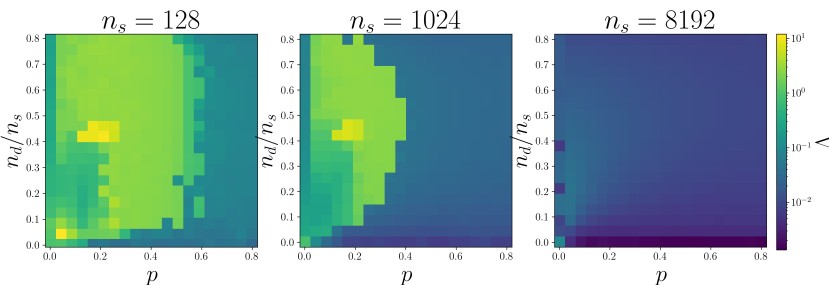

Figure 8: Interference Gaussianity in the large $n_s$ limit. Models are trained at different $p$ and $n_d/n_s$ values. We see that for small $n_s$ only models trained at large $p$ have nearly-uniform off-diagonal entries whereas all models approach uniformity at large $n_s$.

We also plot the maximum absolute difference of the parameters we claim become constant,

$$\Delta_2 \boldsymbol{v} := \max_i \boldsymbol{v}(i) - \min_i \boldsymbol{v}(i).$$

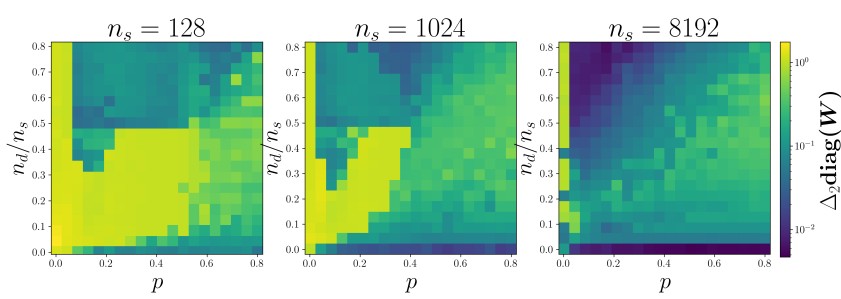

Figure 9: Figure 4 but with $\Delta_2$ instead of $\Delta_1$.

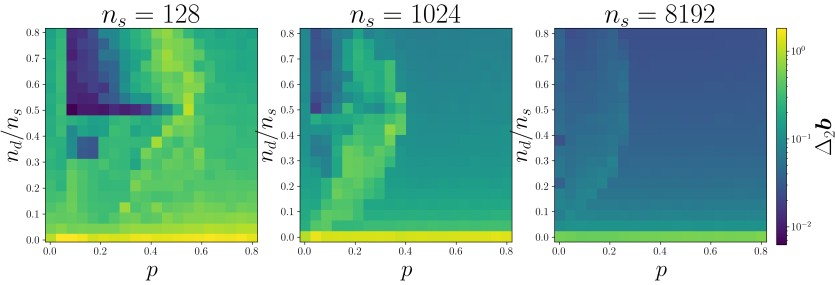

Figure 10: Figure 5 but with $\Delta_2$ instead of $\Delta_1$.

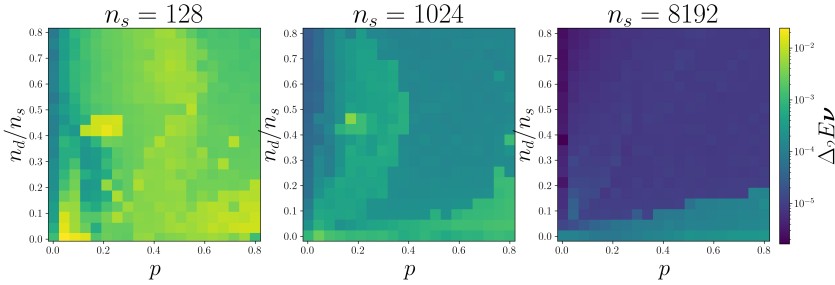

Figure 11: Figure 6 but with $\Delta_2$ instead of $\Delta_1$.

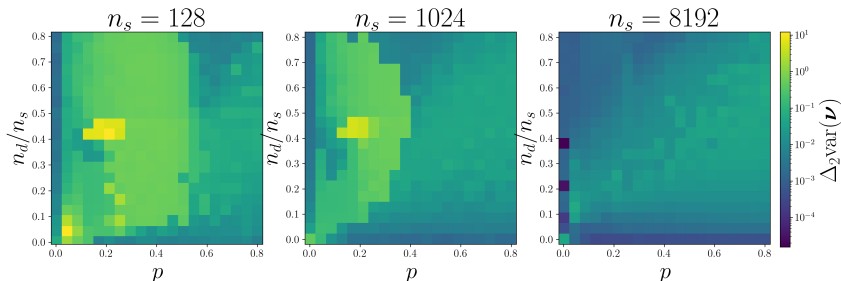

Figure 12: Figure 7 but with $\Delta_2$ instead of $\Delta_1$.

## B  OPTIMIZING OVER $a$

We may optimize over $a$ analytically because it appears almost quadratically in the loss. Consider the expression for the average loss from eq. (5) with $b$ replaced with $a \cdot b$. As long as $a \neq 0$ this redefinition doesn't change the set of accessible models.

Furthermore let us restrict to positive $a$ which allows us to rewrite the loss as

$$L = \mathbb{E}_{x,\nu}[\left(x - \text{ReLU}\left(a(x + \nu + b)\right)\right)^2] = \mathbb{E}_{x,\nu}[\left(x - a\,\text{ReLU}\left(x + \nu + b\right)\right)^2]. \tag{11}$$

The restriction to positive $a$ is acceptable because we never see negative off-diagonal elements in our trained models. Now, optimizing over $a$ is exactly linear regression; we can obtain the optimal value of $a$ with the standard method

$$0 = \frac{d}{da}L = -2\mathbb{E}\left[(x - a\,\text{ReLU}(x + \nu + b))\,\text{ReLU}(x + \nu + b)\right] \tag{12}$$

which implies that

$$a_{\text{opt}} = \frac{\mathbb{E}_{x,\nu}\left[x\,\text{ReLU}(x + \nu + b)\right]}{\mathbb{E}_{x,\nu}\left[\text{ReLU}(x + \nu + b)^2\right]}. \tag{13}$$

Notice that the optimal $a$ is always positive, which is consistent with the assumption we made earlier.

## C  BOUNDING THE RECONSTRUCTION ERROR

**On term:**

We can start to write the on term similarly as

$$\langle(u - \text{ReLU}\left(u + \nu\right))^2\rangle = \left\langle \sigma^2 \int_{-\infty}^{\infty} d\nu \frac{e^{-\frac{(\nu + \frac{|b|}{\sigma})^2}{2}}}{\sqrt{2\pi}}(u - \text{ReLU}(u + \sigma\nu))^2 \right\rangle_u.$$

Now we write the integral in two parts to get rid of the ReLU: one when $u + \sigma\nu < 0$ and one when $u + \sigma\nu > 0$. This gives

$$\underbrace{\left\langle \int_{-\infty}^{-\frac{u}{\sigma}} d\nu \frac{e^{-\frac{(\nu + \frac{|b|}{\sigma})^2}{2}}}{\sqrt{2\pi}} u^2 \right\rangle_u}_{E_r} + \underbrace{\left\langle \sigma^2 \int_{-\frac{u}{\sigma}}^{\infty} d\nu \frac{e^{-\frac{(\nu + \frac{|b|}{\sigma})^2}{2}}}{\sqrt{2\pi}} \nu^2 \right\rangle_u}_{E_\nu}.$$

Where the first term $E_r$ represents error coming from the ReLU and the second term $E_\nu$ represents error coming from the noise. The scaling of $E_\nu$ can be easily bounded:

$$E_\nu < \sigma^2 \int_{-\infty}^{\infty} d\nu \frac{e^{-\frac{(\nu + \frac{|b|}{\sigma})^2}{2}}}{\sqrt{2\pi}} \nu^2 \sim O(\sigma^2 + \sigma b + b^2)$$

And thus we see, unsurprisingly, that we need to set $b \ll 1$ to get a good bound.

To upper bound $E_r$ write the $u$ integral in two intervals: $[0, 2|b|]$ and $[2|b|, 1]$, corresponding to regions in which the interval of the $\nu$ integral does and "decisively" does not include the the mean respectively. In particular, we have

$$E_r < \underbrace{\int_0^{2|b|} u^2 du \int_{-\infty}^{-\frac{u}{\sigma}} d\nu \frac{e^{-\frac{(\nu + \frac{|b|}{\sigma})^2}{2}}}{\sqrt{2\pi}}}_{E_r^{\text{mean}}} + \underbrace{\int_{2|b|}^1 u^2 du \int_{-\infty}^{-\frac{u}{\sigma}} d\nu \frac{e^{-\frac{(\nu + \frac{|b|}{\sigma})^2}{2}}}{\sqrt{2\pi}}}_{E_r^{\text{tail}}}.$$

Since in $E_r^{\text{mean}}$ the $\nu$ integrals' interval includes the mean we may as well extend the interval to the full real line to get the bound, giving

$$E_r^{\text{mean}} < \int_0^{2|b|} u^2 du = O(b^3).$$

$E_r^{\text{tail}}$ can be upper bounded by setting the $\nu$ range to the maximum value of $2|b|$, so we have

$$E_r^{\text{tail}} < (1 - 2|b|) \int_{-\infty}^{-\frac{2|b|}{\sigma}} d\nu \frac{e^{-\frac{(\nu + \frac{|b|}{\sigma})^2}{2}}}{\sqrt{2\pi}} = (1 - 2|b|) \int_{-\infty}^0 d\nu \frac{e^{-\frac{(\nu - \frac{|b|}{\sigma})^2}{2}}}{\sqrt{2\pi}} < O(1)(1 - 2|b|)e^{-\frac{|b|^2}{2\sigma^2}}.$$

Putting it all together gives

$$L < (1 - p)O(\sigma^2 e^{-\frac{b^2}{2\sigma^2}}) + pO(b^3 + e^{-\frac{|b|^2}{2\sigma^2}} + be^{-\frac{|b|^2}{2\sigma^2}} + \sigma^2 + \sigma b + b^2).$$

Plugging in the $b$ scaling from eq. (23) and keeping only the lowest order terms gives

$$L < O(\sigma^2 p \log \frac{1}{p}) \sim O(\frac{p^2}{r} \log \frac{1}{p}). \tag{14}$$

## D  MINIMAL VARIANCE BOUND

We will show a minimum variance bound for matrices $W$ which have all diagonals equal to 1 and also have maximum rank $n_d$. In this case we know that $\text{Tr}\, W = n_s$. On the other hand we also know that the trace is the sum of the eigenvalues, and because $W$ has rank at most $n_d$ that

$$n_s = \sum_{i=1}^{n_d} \lambda_i \tag{15}$$

for the eigenvalues $\lambda_i$ of $W$. Now we solve for the mean of the variance across rows,

$$\frac{1}{n_s} \sum_{i=1}^{n_s} \text{Var}(\nu_i) = \frac{4p - 3p^2}{12 n_s} \sum_{i=1}^{n_s} \sum_{j=1, j \neq i}^{n_s} W_{ij}^2 = \frac{4p - 3p^2}{12 n_s} \left( \text{Tr}(WW^\dagger) - n_s \right). \tag{16}$$

Here the first equality arises from the definition of $\nu_i$ (remembering that we have set the diagonals to 1 exactly) and substituting the variance of $x_j$, while the second equality follows because $\text{Tr}(WW^\dagger)$ is the sum of the square of all entries of $W$, and we subtract off the diagonal entries.

Because we want a bound on this quantity related to the eigenvalues of $W$, it is convenient to use the Schur decomposition of $W = QUQ^\dagger$. Here $Q$ is a unitary matrix and $U$ is upper-triangular with the eigenvalues of $W$ on the diagonal. This allows us to lower bound the trace

$$\text{Tr}(WW^\dagger) = \text{Tr}(QUQ^\dagger QU^\dagger Q^\dagger) = \text{Tr}(UU^\dagger) = \sum_{i,j=1}^{n_s} |U_{ij}|^2 \geq \sum_{i=1}^{n_d} |\lambda_i|^2 \geq \frac{n_s^2}{n_d} \tag{17}$$

where the last inequality follows from Cauchy-Schwarz and eq. (15). With this we find a bound on the variance

$$\frac{1}{n_s} \sum_{i=1}^{n_s} \text{Var}(\nu_i) \geq \frac{4p - 3p^2}{12} \left( \frac{n_s}{n_d} - 1 \right), \tag{18}$$

with equality if $W$ is symmetric with all non-zero eigenvalues equal. These two conditions follow because the two inequalities in the proof become equalities when these conditions are met. This naturally leads to a candidate for the optimal choice of $W$, namely matrices of the form

$$W \propto OPO^T \text{ and } W_{ii} = 1 \tag{19}$$

where $O$ is an orthogonal matrix and $P$ is any rank-$n_d$ projection matrix. This kind of matrix saturates both bounds because it is symmetric and has all nonzero eigenvalues equal to 1.

# E    BOUNDS ON THE LOSS

## E.1    UPPER BOUND ON LOSS SCALING

We now show that the loss drops off quickly in the sense that for $\frac{r}{p} \gg 1$ we get that $L(p)/p \to 0$, i.e. $L(p)$ scales super-linearly with $p$. We will consider the regime where $r \ll 1$ holds[3] so that we may take $\sigma^2 \sim \frac{p}{r} \ll 1$.

To obtain the upper bound we will make educated estimates for values of $a$ and $b$ that are near optimal. In particular, in appendix B we show that the optimal value of $a$ is (after absorbing the mean of $\nu$ into $b$):

$$a_{\text{opt}} = \frac{\mathbb{E}_{x,\nu}\left[x\,\text{ReLU}(x+\nu)\right]}{\mathbb{E}_{x,\nu}\left[\text{ReLU}(x+\nu)^2\right]}. \tag{20}$$

From the form of the loss, we know that $b$ must decrease as $p$ decreases for the loss to go down faster than $O(p)$. Thus $\nu$ has both a mean and variance approaching 0, and $a_{\text{opt}} \to 1$. Thus we plug in $a = 1$ before taking these limits in the expectation of getting a good upper bound. The loss then takes the form

$$L = (1-p)L_{\text{off}} + pL_{\text{on}}$$

with

$$L_{\text{off}} = \mathbb{E}\left[\text{ReLU}\left(\nu\right)^2\right], \text{ and}$$

$$L_{\text{on}} = \mathbb{E}\left[\left(u - \text{ReLU}\left(u+\nu\right)\right)^2\right]$$

**Off term:**    The off term can be upper bounded via

$$\mathbb{E}\left[\text{ReLU}(\nu)^2\right] = \int_0^\infty d\nu \frac{e^{-\frac{(\nu+|b|)^2}{2\sigma^2}}}{\sqrt{2\pi\sigma^2}}\nu^2 = \sigma^2 \int_0^\infty d\nu \frac{e^{-\frac{(\nu+\frac{|b|}{\sigma})^2}{2}}}{\sqrt{2\pi}}\nu^2 \tag{21}$$

$$\leq \frac{\sigma^2}{2}e^{-\frac{b^2}{2\sigma^2}} \tag{22}$$

and thus we see we need to set $\frac{b}{\sigma} \gg 1$ to get a good bound. In particular, we know empirically that the loss drop happens at increasingly smaller $r$. To ensure this we let $\sigma^2 \sim \frac{p}{r}$ scale at some rate slower than $p$. Thus to ensure that the total loss decreases faster than $O(p)$, we need $e^{-\frac{b^2}{2\sigma^2}} \sim O(p)$ or in other words

$$b \sim \sigma\sqrt{\log\frac{1}{p}}. \tag{23}$$

**On term:**    We perform a similar, but slightly more involved computation in appendix C and combine with the off term to obtain

$$L < (1-p)O(\sigma^2 e^{-\frac{b^2}{2\sigma^2}}) + pO(b^3 + e^{-\frac{|b|^2}{2\sigma^2}} + be^{-\frac{|b|^2}{2\sigma^2}} + \sigma^2 + \sigma b + b^2).$$

Plugging in the $b$ scaling from eq. (23) and keeping only the lower order terms gives

$$L < O\left(\sigma^2 p \log\frac{1}{p}\right) \sim O\left(\frac{p^2}{r}\log\frac{1}{p}\right). \tag{24}$$

## E.2    LOWER BOUND ON LOSS SCALING

We now show a lower bound on the loss in the $p \to 0$ limit. To do this, we will show a more general lower bound on the on loss for any deterministic function of the pre-activation. Specifically, we would like to lower bound

---

[3]For example $r = p^{1-\epsilon}$ for any $\epsilon \in (0,1)$.

$$L \leq pL_{\text{on}} = \mathbb{E}[(u - f(u + \nu))^2]$$

for any function $f$ with $u \sim \text{Uniform}[0, 1]$ and $\nu \sim \mathcal{N}(0, \sigma)$. Recall that there is no need to consider the bias $b$ as it can be absorbed into $\nu$. Recall that the optimal function $f$ is given by

$$f^*(u + \nu) = \mathbb{E}[u|u + \nu].$$

Let's make a change of variable from $u, \nu$ to $y \equiv u + \nu, u$, and then use the tower property to rewrite $L_{\text{on}}$ as

$$L_{\text{on}} = \mathbb{E}_{y \sim P_{u+\nu}} \left[ \mathbb{E}_{u|y} \left[ (u - f^*(u + \nu))^2 \right] \right]. \tag{25}$$

We first draw $y$ from the marginal distribution of $u + \nu$ and then draw $u$ from the conditional distribution given $y$. Because $f^*$ is exactly the conditional expectation the interior expectation becomes the conditional variance

$$L_{\text{on}} = \mathbb{E}_y \left[ \text{Var} \left[ u|y \right] \right]. \tag{26}$$

Because we want to lower bound $L_{\text{on}}$ it will be convenient to start with a lower bound for the conditional variance. We will lower bound the conditional variance for $y \in [\sigma, 1 - \sigma]$, and then use that lower bound to find a lower bound for the loss, with a goal of showing that the loss is lower bounded by a constant multiple of $\sigma^2$, for $\sigma < \frac{1}{4}$. This will show that the overall loss of any strategy, even one which can perfectly estimate which features are on or off, is incapable of achieving a reconstruction error better than $O(p^2/r)$.

The conditional distribution for $u$ is a truncated Gaussian distribution. By Bayes' theorem

$$P[u|u + \nu = y] = \frac{P[u + \nu = y]P[u]}{P[y]} \tag{27}$$

$$= \begin{cases} \frac{e^{-(u-y)^2/2\sigma^2}}{\int_0^1 dx e^{-(x-y)^2/2\sigma^2}} & \text{if } u \in [0, 1] \\ 0 & \text{otherwise,} \end{cases} \tag{28}$$

with normalizing constant $Z(y) = \int_0^1 dx e^{-(x-y)^2/2\sigma^2} < \sqrt{2\pi\sigma^2}$. This is a truncated Gaussian distribution. Fix $y \in [\sigma, 1 - \sigma]$ so that all distributions are implicitly conditioned on $y$ for now. Sample $u$ via the following procedure. First we decide if $|u - y| \leq \sigma$ and then we either sample from the conditional distribution $P[u|y$ and $|u - y| \leq \sigma]$ or $P[u|y$ and $|u - y| \geq \sigma]$ with their corresponding probabilities. Let $R$ be the indicator random variable denoting $|u - y| \leq \sigma$. Then by the law of total variance

$$\text{Var} \left[ u \mid y \right] = P[R = 1] \text{Var} \left[ u \mid R = 1 \right] + P[R = 0] \text{Var} \left[ u \mid R = 0 \right] + \text{Var}_R \left[ \mathbb{E}[u \mid R] \right] \tag{29}$$

$$\geq P[R = 1] \text{Var} \left[ u \mid R = 1 \right] \tag{30}$$

where we have dropped the latter two positive terms to derive the lower bound. $P[R = 1] \geq \text{erf}(2^{-1/2})$ because the chance a truncated Gaussian is within one $\sigma$ of its mode is larger than that for an untruncated Gaussian, given that the truncation is more than $\sigma$ away from the mode. This condition is satisfied by construction because we have chosen $y$ to be more than $\sigma$ from the boundary.

Additionally a trivial scaling argument shows that the variance is proportional to $\sigma^2$ which means that there is some constant, $C > 0$ such that

$$\text{Var} \left[ u \mid y \right] \geq C\sigma^2 \tag{31}$$

when $y \in [\sigma, 1 - \sigma]$. To complete the argument we now return to

$$L_{\text{on}} = \mathbb{E}_y \left[ \text{Var} \left[ u|y \right] \right] \geq \mathbb{E}_y \left[ \text{Var} \left[ u|y \right] 1_{y \in [\sigma, 1 - \sigma]} \right] \geq C\sigma^2 P[y \in [\sigma, 1 - \sigma]]. \tag{32}$$

For $\sigma = 1/4$ this probability is clearly finite and for $\sigma < 1/4$ it is increasing as $\sigma$ decreases so it is uniformly bounded below by a constant $C_1$. So finally

$$L_{\text{on}} \geq C'\sigma^2 \implies L \geq C'p\sigma^2 \approx \frac{C'p^2}{r}. \tag{33}$$

# F  PERSIAN RUG CONSTRUCTION SATISFIES PERMUTATION SYMMETRY

In this section we provide a short discussion on Hadamard matrices, and more importantly proofs that our Persian Rug construction satisfies the permutation symmetry conditions.

A $n \times n$ matrix $H$ is a Hadamard matrix if every entry of $H$ is either $1$ or $-1$, and if all the rows of $H$ are orthogonal. This implies that $HH^T = nI$ where $I$ is the identity matrix.

As a reminder we construct the rug matrix of rank $n_d$ and size $n_s$ by first choosing a subset $S \subseteq \{1, \ldots, n_s\}$ of size $|S| = n_d$. Then we construct

$$R_{ij} = \frac{1}{n_d} \sum_{k \in S} H_{ik} H_{jk} \tag{34}$$

for any Hadamard matrix $H$ of dimension $n_s$

These properties are sufficient for us to prove the required symmetries, as well as the spectral properties of $R$:

- $R_{ii} = 1$,

- For any $i = 1, \ldots n_s$ that $\sum_{j=1, j \neq i}^{n_s} R_{ij}^2 = \frac{n_s}{n_d} - 1$,

- $R$ is proportional to a projector.

The first property is apparent from the fact that all entries of $H$ are $\pm 1$.

$$R_{ii} = \frac{1}{n_d} \sum_{k \in S} (H_{ik})^2 = \frac{1}{n_d} \sum_{k \in S} (\pm 1)^2 = 1. \tag{35}$$

The second property follows similarly, but with some more algebra. Without loss of generality let $i = 1$ so that we consider the first row's off-diagonal terms and let $\delta_{ij}$ denote the Kronecker delta symbol. Then their sum is

$$\sum_{j=2}^{n_s} R_{1j}^2 = \frac{1}{n_d^2} \sum_{j=2}^{n_s} \sum_{k_1, k_2 \in S} H_{1k_1} H_{1k_2} H_{jk_1} H_{jk_2} \tag{36}$$

$$= \frac{1}{n_d^2} \sum_{k_1, k_2 \in S} H_{1k_1} H_{1k_2} \sum_{j=2}^{n_s} H_{jk_1} H_{jk_2} \tag{37}$$

$$= \frac{1}{n_d^2} \sum_{k_1, k_2 \in S} H_{1k_1} H_{1k_2} (H_{\cdot k_1} \cdot H_{\cdot k_2} - H_{1k_1} H_{1k_2}) \tag{38}$$

where we use the notation $H_{\cdot k}$ for the $k^{\text{th}}$ row of the matrix $H$ viewed as a vector. We know these rows are orthogonal and have norm $n_s$ because all their entries are $\pm 1$ so

$$\sum_{j=2}^{n_s} R_{1j}^2 = \frac{1}{n_d^2} \sum_{k_1, k_2 \in S} H_{1k_1} H_{1k_2} (n_s \delta_{k_1 k_2} - H_{1k_1} H_{1k_2}) \tag{39}$$

$$= \frac{n_s}{n_d^2} \sum_{k_1, k_2 \in S} H_{1k_1} H_{1k_2} \delta_{k_1 k_2} - \frac{1}{n_d^2} \sum_{k_1, k_2 \in S} H_{1k_1} H_{1k_2} H_{1k_1} H_{1k_2} \tag{40}$$

$$= \frac{n_s}{n_d^2} \sum_{k \in S} H_{1k}^2 - \frac{1}{n_d^2} \sum_{k_1, k_2 \in S} H_{1k_1}^2 H_{1k_2}^2 = \frac{n_s}{n_d} - 1, \tag{41}$$

where we used the fact $|S| = n_d$ and the unit norm of the entries to simplify. This shows that the noise $\nu_i$ has the same variance over all rows.

Finally we show that $R$ is proportional to a projector. Looking at

$$(R^2)_{ij} = \frac{1}{n_d^2} \sum_{l=1}^{n_s} R_{il} R_{lj} \tag{42}$$

$$= \frac{1}{n_d^2} \sum_{k_1,k_2 \in S} \sum_{l=1}^{n_s} H_{ik_1} H_{lk_1} H_{lk_2} H_{jk_2} \tag{43}$$

$$= \frac{1}{n_d} \sum_{k_1,k_2 \in S} H_{ik_1} \delta_{k_1 k_2} H_{jk_2} \tag{44}$$

$$= \frac{1}{n_d} \sum_{k \in S} H_{ik} H_{jk} = R_{ij} \tag{45}$$

The fact $R^2 = R$ means that $R$ is a projector, and hence it is proportional to a projector.

# G  TRAINING DETAILS

For all of the toy models (eq. (1) with $n_s \in \{128, 1024, 8192\}$) and use a learning rate of $3 \times 10^{-3}/\sqrt{n_s}$. For the Hadamard model ($n_s = 8192$), we use the same stopping strategy described for the toy models. We use a batch size of 512, maximum number of epochs of 100, a learning rate of $3 \times 10^{-1}/\sqrt{n_s}$, and also train 5 models and keep the model with the lowest loss.

# H  PARTIALLY BREAKING PERMUTATION SYMMETRY

A natural question following our analysis; To what extent do the qualitative features of the results we derive depend on the permutation symmetry of the input vectors $\mathbf{x}$? To move away from this assumption a little, we perform numerical experiments on the following loss

$$L = (\mathbf{x}; W_{\text{out}}, W_{\text{in}}, \mathbf{b}) = n_s^{-1} \sum_{i=1}^{n_s} M_i (x_i - f_{\text{nonlinear}}(\mathbf{x})_i)^2 \tag{46}$$

where $M_i$ are weights which control the importance of each feature. We choose

$$M_i = \begin{cases} 1 & \text{if } i \le \frac{n_s}{2} \\ \alpha & \text{if } i > \frac{n_s}{2} \end{cases} \tag{47}$$

for some parameter $\alpha \in [0,1]$. This breaks the symmetry because some features are now more important than other features. As before we train until the loss function ceases to decrease, with a batch size of 4096 and a learning rate of .0003. We train a model with $n_s = 4096$ sparse features which have the same $p = .04$ of activating. The compressed dimension $n_d = 1024$.

First let us check that despite the symmetry breaking all features are still represented. Looking at the model with $\alpha = \frac{1}{2}$ we can consider the diagonal and off-diagonal terms. The diagonal terms can be broken up into two groups: $W_{ii}$ for $i \le n_s/2$ which have high importance, and $W_{ii}$ for $i > n_s/2$ which have lower importance. We plot a histogram of these terms in the first panel of fig. 13. The terms within each group are close to uniform whereas the two groups have somewhat different means with the important features having slightly larger diagonal entries.

The off-diagonal terms can be summarized by the standard deviations of $\nu_i$ as before, again split into more and less important groups. We see that the variance of $\nu_i$ is almost uniform inside each group, and smaller for the more important features. This shows that permutation symmetry is maintained within each group, all features may still be represented, and hints that the model gracefully deviates from our permutation-symmetric solution by shifting it's budget for the noise and signal.

To see that this shift behaves nicely as $\alpha$ varies away from 1 (which recovers full permutation symmetry), we look at the mean diagonal value in each group (left panel of fig. 14), and the mean standard deviation of the noise $\nu_i$ (right panel of fig. 14) in each group. As we can see for $\alpha$ near

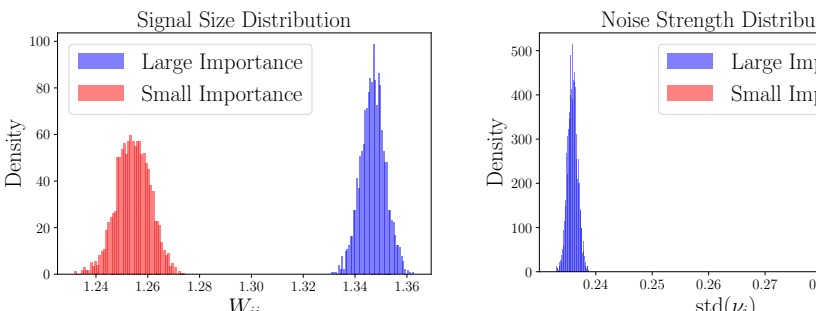

Figure 13: Distributions of diagonal terms (of $W$) in a single trained model with $n_s = 4096$ sparse features, $n_d = 1024$ dense dimensions, and relative importance weight $\alpha = 1/2$ for the less important features. The first subfigure shows that the distribution of diagonal components with small importance (red) and large importance (blue) are separated, but similar in magnitude. On the other hand The distribution of the noises is different, with more noise allocated to the less important features.

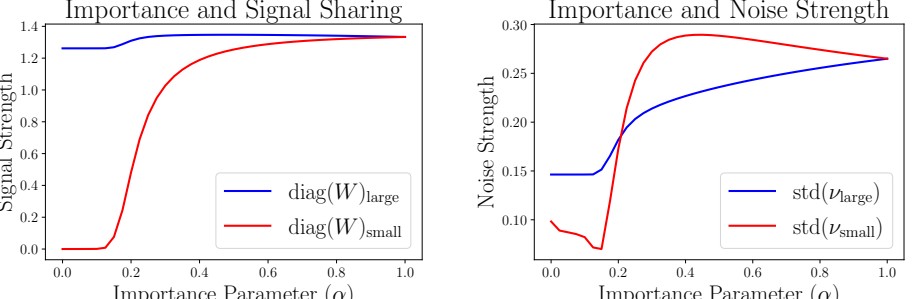

Figure 14: This figure shows how the signal strength (left) and noise level (right) are shared between the more and less important features as the importance parameter $\alpha$ goes from 0 to 1. The signal strength is given by the mean diagonal value of $W$, while the noise is given by the mean over rows of the standard deviation of $\nu_i$. When $\alpha$ is small all the capacity of the model is directed towards the more important features. As $\alpha$ increases the model begins to dedicate some capacity towards the less important features. At this point the model pushes more of the noise towards the less important features. The model breaks symmetry smoothly near $\alpha = 1$.

1 the behavior shown in the histograms is maintained. As $\alpha$ becomes smaller the model initially decreases the signal strength, and increases the noise associated with the less important group of sparse features.

Around $\alpha \approx .25$ (for this set of parameters) the model begins to give up entirely on encoding the less important features, which allows it to increase the fidelity of the more important group. It does this primarily by reducing the noise sent to those features.

Here we see that the deviations from permutation symmetry produce slowly varying changes in the optimal encoding strategy for a wide range of $\alpha$. This implies that qualitative features of the permutation symmetric setting may remain, even when this symmetry is broken in a more realistic setting.

# I   CONFIRMING THE BOUNDS EMPIRICALLY

In this section, we confirm that the bounds we derive agree empirically with that of the numerically trained Hadamard models. The details of the parameters are given as follows:

- $n_s = 32,768$
- $r$ ranges from 0.01 to 0.1 uniformly (10 steps)
- $p$ ranges from 0.0001 to 0.01 (10 steps)
- We keep all triplets of the form $(n_s, r, p)$ where $pn_s > 5$ and $p < 0.1r$ and ignore the rest.

We then train Persian Rug models with these parameters. We plot the final loss of these models against $p^2/r$ in fig. 15.

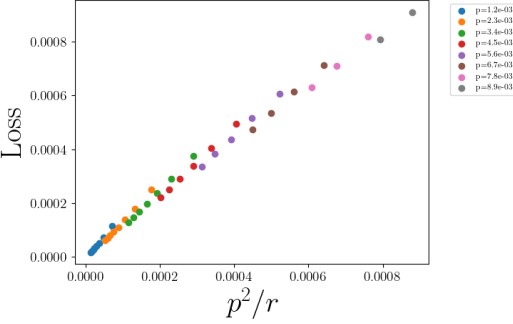

Figure 15: Here, we choose parameters to capture the regimes $p, r << 1$, $p << r$, $pn_s >> 1$ and plotted the loss of the Hadamard model against $p^2/r$. The linear relationship with a slightly decreasing slope with $p$ suggests the $p^2 \log(1/p)/r$ bound is correct.