# OpenReview forum: "The Persian Rug: solving toy models of superposition using large-scale symmetries"
_ICLR.cc/2025/Conference — Submitted to ICLR 2025_

### Official Review · Reviewer_PspC · 2024-10-31

**Soundness:** 3
**Presentation:** 2
**Contribution:** 3
**Rating:** 6
**Confidence:** 3

**Summary:**

This paper examines a toy model of superposition, revealing that when the input data exhibits permutation symmetry, the auto-encoder learns an algorithm responsive only to macroscopic parameters. Using this insight, the authors derive upper bounds on the reconstruction loss and design hand-crafted symmetric models that match the loss of the trained model.

**Strengths:**

- This paper presents a novel study on the effects of input symmetry in superposition models.
- This paper identifies a statistical permutation symmetry in the learned model weights, which is verified by designing a "Persian rug model" that achieves similar performance to the trained model.
- The authors provide theoretical upper bounds for the reconstruction loss, which their experiments confirm by showing a notable drop in loss around $r\approx p$.
- Detailed explanations throughout the paper improve the understanding and interpretation of the experimental results.

**Weaknesses:**

- **Writing**: The empirical observations in the paper feel overly repetitive, while the theoretical sections come across as overly technical and challenging to follow. For example, Section 3 spends three pages describing that the diagonal elements and bias become uniform while off-diagonal elements become noisy. Section 4 includes excessive technical detail in the derivations, which could be condensed into formal theorems, with the technical details moved to the appendix for easier readability.

- **Theory**: Many theoretical discussions in this paper lack rigor. For instance, the discussion between lines 353 and 361 is confusing and does not seem to follow rigorous mathematical reasoning. I will provide further points of confusion in the questions section. Notably, as a theoretically-focused paper, it lacks any formal theorems, making it difficult to digest. Summarizing results into well-defined theorems with clearly explained notations would improve clarity.

- **Experiments**: The experimental scope appears limited. The paper only presents results on a toy model with input data following a sparse, uniform, permutation-symmetric distribution. However, real-world data, such as natural language, does not exhibit permutation symmetry. Including experiments on real-world data would help demonstrate the applicability of these results.

In summary, the experimental scope of this paper seems narrow, and the theoretical sections lack the rigor expected of a theoretical work. I recommend that the authors refine their presentation of theoretical results and expand their experiments. I’m open to any clarifications if there has been a misunderstanding of the paper's theory from my end.

**Questions:**

- In Figure 1, you show that the structure of the artificial weights resembles a Persian rug. Out of curiosity, what do the trained weights look like?

- On line 155, you mention that in the $r \to \infty$ limit, the diagonal elements of $\mathbf{W}$ become the same, while the off-diagonal elements become zero-mean noise, resulting in $\mathbf{W} = a \mathbf{I}$. However, since $\mathbf{W}$ is the product of two low-rank matrices, shouldn’t it have a rank no greater than $n_d$? How, then, does it approach a full-rank matrix?

- On line 355, the symbol $\mu$ is not defined. Could you also clarify the main message of this paragraph?

- On line 406, the parity function is not defined.

- On line 410, should the LHS be $R_{ij}$? It would also be helpful to include a theorem stating that $\mathbf{R}$ satisfies the statistical symmetries, with a proof deferred to the appendix.

---

> ### Author Response · Authors · 2024-11-24
> **Official Comment by Authors**
>
> We thank the reviewer for their careful review and address the weaknesses they brought up as well as their questions here.
>  ## Weaknesses
> **Writing:** We agree with the reviewer about the repetitiveness of the empirical observations and have moved the majority of the figures to the appendix. We have also removed unnecessary technical details in section 4. However we believe our informal style is appropriate for this type of empirically backed and exploratory theoretical work, as we believe the function of formalization is primarily to avoid errors or to enshrine well-established results.
>
> **Theory:** We have addressed the reviewer's concerns regarding the confusing wording of this paragraph. Additionally, we emphasize Section 4.1 is meant to give a qualitative description of the algorithm before rigorous arguments are given in later sections.
>
> **Experiments:** Our work demonstrates that models become easier to analytically analyze when their weights inherit symmetries in their training data. While we considered the maximal case of permutation symmetry, a similar simplification of the loss occurs if the weights inherit smaller symmetries. As a demonstration we have added\inc{ \textbf{look at appendix}. } Real world data is of course significantly more complicated, but we nevertheless expect model weights to reflect its structure. However, we believe a first principles understanding of how this occurs requires first the development of fundamental tools to analyze toy models of increasing sophistication. You may also see [this discussion](https://www.alignmentforum.org/posts/uvEyizLAGykH8LwMx/fundamental-vs-applied-mechanistic-interpretability-research) on the differences between fundamental and applied mechanistic interpretability.
>
> ## Questions
> **Q1:** An example of the trained weights can be seen in figure 3.
>
> **Q2:** The size of the off-diagonal noise becomes zero element-wise in the limit, but does not go to zero averaged over a row. Therefore it is not appropriate to write that $W = aI$ in the limit. In fact, the off-diagonal elements encode enough correlations so that $W$ is indeed low-rank. (Note also that $r\to\infty$ was a typo, we meant $n_s\to\infty$ with $r$ fixed.)
>
> **Q3:** $\mu$ referred to the mean value of the noise $\nu$. We have clarified the exposition by absorbing it into the bias $b$. The goal of the paragraph is to explain the context and give intuition for the calculation that follows, and we have edited it to improve the clarity.
>
> **Q4,5:** We have more clearly defined a Hadamard matrix by its defining properties, which makes it more clear how the construction works. We have also included a new section in the appendix explaining why it satisfies the permutation symmetry for off-diagonal terms along with a step-by-step discussion of the other two conditions. Additionally, we fixed the $R$ versus $R_{ij}$ typo.

---

> > ### Comment · Reviewer_PspC · 2024-11-25
> > **Response to Authors**
> >
> > I thank the authors for the response and for addressing my questions. Although I still have some concerns about the paper's readability and writing style, the authors have addressed my theoretical concerns, and I have adjusted my score accordingly. However, as I have not thoroughly verified all the theoretical details, I recommend considering the more detailed evaluations provided by other reviewers.

---

### Official Review · Reviewer_dnUr · 2024-10-31

**Soundness:** 2
**Presentation:** 2
**Contribution:** 1
**Rating:** 3
**Confidence:** 4

**Summary:**

The paper studies the following auto-encoder model: linear encoding + linear decoding with additional denoising (shifted-ReLU) application on top. This model is applied in context of compression of i.i.d. sparse uniform data. The authors provide an analysis of the model behaviour (MSE bound) and identify the set of matrices which matches the performance of the trained model.

**Strengths:**

The paper provides to some extent a solid analysis of the proposed model.

**Weaknesses:**

The whole interpretability premise is evidently far fetched especially whenever LLMs appear in the related context. The paper does not provide any evidence that the phenomenology observed in their i.i.d. + small model setting appears on any even "toy"-ish real data, or justify enough whether the prescribed methodology could be put forward to gain an understanding of the behaviour of the real world models. The suggestion is to reframe and focus on the subject of substance - analysis of the autoencoder model.

I fail to see why so much of the paper space is occupied by a the "Gaussian-ity" experiments in Section 3.2. The authors do no provide any rigorous training results anyway (that the model will converge to a specific subset of matrices). In this view, since the proposed behaviour is assumed anyway using the empirical evidence, the amount of histograms and the accompanying description is clearly an overshot (which does not really add any further weight to the submission).

The technical machinery used for the analysis is not of a separate interest.

The main concern, however, is the emergence of a certain structure in the weights. In particular, it is completely unclear whether the "Persian carpet" structure is indeed favoured by the network implicitly or just a set of weights which achieves the close to optimal MSE.
If later, the observations in the paper does not shed any light on any sort of "interpretability", since the network does not implement the proposed weight pattern when trained. Prior, however, is not supported by any evidence (no SGD/gradient flow convergence result).

The phenomena of uniform enough weights is not surprising on its own given that the signal is i.i.d., especially in the view of related work.
Specifically, could authors provide any intuition why specifically "Persian carpet" (except visual appeal)? There is no evidence whatsoever that a simpler uniform design will not achieve desired. Why not use some other design: for instance, rotationally invariant matricies with specific spectra (if the input distribution is centered, see next paragraph) and exploit Approximate Message Passing analysis (granted, the optimal denoising will not be close to ReLU)?

Another concern: why the data is not centered (which is *common practice*)? It is hard to judge, in this case, how much of an effect on the proposed structure it has, in the view of ReLU - which has cut-off at zero.

I also assume that in equation (2) there is normalization in dimension missing. Otherwise, dropping "i" in (5) does not make sense.

Lastly, the particular choice of ReLU model (1) in the view of above remains quite unclear. It is hard to judge if any of the observations are not an artifact of the modeling.

The authors fail to acknowledge *a lot* of the related literature which is non-trivially connected to the observed model behaviour. Below, a non-exhaustive list:
-  AEs: (Refinetti & Goldt, 2022); (Cui & Zdeborova, 2023); "Analysis of feature learning in weight-tied autoencoders via the mean field lens" Nguyen (2021); (Shevchenko et al., 2023); (Kögler et al., 2024);
- Approximate Message Passing algorithms: (Donoho et al., 2009); rotationally invariant design (Rangan et al., 2019; Schniter et al., 2016; Ma & Ping, 2017; Takeuchi, 2019); optimal spectral design (Ma et al., 2021).

**Questions:**

See the weaknesses section.

---

> ### Author Response · Authors · 2024-11-24
> **Official Comment by Authors**
>
> We thank the reviewer for their review and their insight into connections with the literature. Here we respond to the weaknesses brought up.
>
> **Interpretability and Autoencoders:**
> It is a fair point that the main substance of the paper is the analysis of the autoencoder model. But this autoencoder model was put forth as a toy model of superposition, which is an important concept in the science of interpretabability. This is why we discuss the implications of our results to interpretability in the abstract, introduction, and conclusion. If there are any specific sections or sentences you think should be removed or further qualified, please let us know.
>
> **Main Concern:** We agree with the reviewer that the trained neural networks do not converge to the Persian Rug and indeed we tried to emphasize this in the paper. So we are in the "latter'' case mentioned by the reviewer. However, our work does in fact interpret the learned algorithm as much as possible because only certain statistics of the weights are important. The Persian Rug has additional structure, which we warn against interpreting. While this is just for the toy model of compressing independent features, we think this meta point on mechansitic interpretability is worth emphasizing for future researchers wanting to do circuit analysis. We give the exact conditions as conditions on the spectrum and the diagonal for optimal $W$, and the Persian Rug as an example.
>
> **Centering of Data:** In this context, assuming the sparse data is positive is standard (see Elhage et al. 2022, Ganguli and Sompolinksy 2010, and also the references in reviewer ru6M's comments on our work).
>
> **Choice of ReLU Nonlinearity:** We have added to the paper a lower bound for the loss with the optimal elementwise activation function. The fact that the upper bound we derive for the ReLU case is equal (up to logarithmic factors) to this lower bound shows that the ReLU belongs to a class of models which all share the same problem with respect to reconstruction -- they don't accurately predict the sparse reconstructed variables. It's therefore a choice to show a general principle.
>
> **Connection to Literature:** We have addressed the referees' concerns regarding the lack of attention towards two parts of the literature mentioned in their review. We have included a discussion of the connection between these subjects and our paper, which was not immediately apparent to us and we thank the referee for bringing them up. The first group of papers primarily deal with autoencoders of type [1] $\hat{x} = A \sigma(Bx)$. On the other hand, we work with autoencoders of the form [2] $\hat{x} = \sigma(ABx)$ where the distribution over $x$ is highly non-Gaussian. In our setting [2], we find that training with $x$ following a Gaussian distribution matching the first two moments does not recover similar $A$ or $B$, for example the overall scale $|AB|$ is significantly different between Gaussian and non-Gaussian $x$. This is unlike setting [1] where Gaussian and non-Gaussian $x$ behave similarly.
>
> Our work also has an oblique relationship with message passing (and convex optimization) algorithms for signal recovery. While these are powerful algorithms for solving the general problem of signal recovery, we are interested in studying the specific recovery algorithm a network learns when it is simultaneously allowed to choose the encoding scheme. The algorithm is constrained by the network's architecture: a 1-layer network which decodes the sparse signal by applying a linear layer followed by a gating. This is similar in structure to a single step of a message passing algorithm (and also similar to [1] from reviewer ru6M). However what our analysis shows is that even in the best case dictionary the network can find, this model cannot efficiently decode the sparse signal without error, which is not the main area of interest in the current literature. Finally, our aim is to speak to the interpretability community, who are widely using the model (or very similar models) to the one we consider, and not to push forward the general theory of autoencoders. The ideas in our manuscript are *not* well known in this area, nor can they be derived from the literature cited by the reviewer.

---

> > ### Comment · Reviewer_dnUr · 2024-11-26
> >
> > I thank the authors for their response. However, I will stand by my original evaluation, especially, given the other reviewers' concerns.

---

> > > ### Author Response · Authors · 2024-11-27
> > > **Request for specific remaining concerns**
> > >
> > > As we have carefully engaged with the reviewer's concerns and questions both with changes to the draft and clarifications to their questions, we respectfully ask the reviewer to specifically state their remaining concerns, especially in light of the large discrepancy in the opinion of the reviewers.

---

### Official Review · Reviewer_iN6D · 2024-10-31

**Soundness:** 3
**Presentation:** 2
**Contribution:** 3
**Rating:** 6
**Confidence:** 3

**Summary:**

The work tries to derive analytical ways to obtain the near-optimal solutions of a specific type of autoencoders for synthetic sparse datasets. After theoretical analysis, it provides the process for constructing the autoencoder’s weight matrix without numerical optimization.

**Strengths:**

1. Has theoretical analysis that looks legit to me, even though it is only focused on a very specific type of autoencoders on a specific type of datasets.
2. The observation leading to theoretical analysis structure of this work can effectively convey the authors’ motivation.

**Weaknesses:**

The main weakness of this paper is its inconsistency and the absence of experiments for supporting their theoretical findings in Section 4, which greatly undermines the soundness of their results. Also, the authors should avoid introducing jargons and verbosity, be concise and accurate about what they mean, or provide necessary references if the jargons are widely accepted elsewhere.
I suggest that the authors spend more time streamlining their work to improve its readability and precision, and bolster their theoretical claims with empirical results.

Regarding contribution, the results in this paper might be too restricted to a specific type of problem and model. I'm not sure how crucial this problem is to LLM development, and because I also value rigorous theoretical studies of problems no matter how small they seem, I refrain from saying harsh words like "it's trivial". That said, the authors had better show their problem's importance with better literature review and critical discussion.

## Major

### Abstract

1. Some claims in the abstract are not (well) reflected in the main text. For instance, “changes to the elementwise activation function or the addition of gating can at best improve its performance by a constant factor”. I guess this is about the discussion on Line 486 to 492. Experiments need to be done to verify this.

### Section 1

1. Line 76: Please provide clear definition of “permutation symmetric data” and “thermodynamic limit”. At least they do not appear in the work (Elhage 2022) you cited, and they sound kinda vague and unintuitive. For instance, how can “permutation symmetric” be derived from “no input feature is privileged”? As long as the features are not identical you cannot say it’s “permutation symmetric or invariant”, right? If these are jargons commonly used in your out-of-ML research area, you need to translate it for your mortal audiences in the ML community.

### Section 2

1. How many samples are there in your dataset on Line 111? What is the rank of this data matrix you sampled? Could the rank of this data matrix affect or even undermine your later observations and motivations? Did you scale up the number of samples as you increase $n_s$? If so, how? You need to provide more details about your experiments.

### Section 3

1. On line 142 there is *“due to an immediate drop in the loss starting around nd/ns ≈p. The slope and duration of this initial fall is controlled by p. In particular, in the high-sparsity regime (pclose to zero), the loss drops to zero entirely near the nd/ns ≈0 regime.”*
I don’t think any part of Fig. 2b can clearly show this, especially considering that the upper bound of the x-axis (0.8) is way larger than p=0.05. You need to plot the loss curve at different p to demonstrate that, and zoom in on the “nd/ns ≈0 regime” and “nd/ns ≈p regime” to illustrate your points. Also, use the `subfigure` environments to enclose the subfigures.
2. On line 152, there is *“nd/ns →∞ limit”*, but Fig. 3,4,5,6 cannot show any of such limit. All nd/ns are under 0.8.  Do you mean “nd/ns →0 limit” (although this makes no sense), or just ns →∞? Because of this, I'm uncertain when reading your statements from Line 155 to 161.
    * If it's the latter, as I understand it, 8192 might still be kinda small.
    * Are you always assuming that nd < ns throughout the entire paper? Be clear.
3. Should provide a clear definition of “permutation invariant” mathematically. Also, what is “permutation symmetric” matrix? Be clear about what you say, or provide citations at least.
4. Elaborate on “We hypothesize this occurs because for small models (and small feature probabilities) the contribution to any output from off-diagonal terms may fluctuate because of the small size of the matrix.” on Line 215. It’s like saying many things without saying anything.
5. Elaborate on Line 236 to 239. Actually, this sentence is too convoluted to read.
6. On Line 236 “Similar features are present in the two plots”. On Line 240 “we give three plots”. What and where are these five plots? Be specific.
7. For Line 266 to 269 and Fig. 5. What is $\nu$? What is $\Delta \text{var}(\nu)$? What is “the fluctuation of the total norm of the off-diagonal elements in each row in W across rows”? Is it simply the variance of the off-diagonal elements’ variances? What do you mean by signal-to-noise ratio? What is considered as noise and what is not? The writing surely needs to be improved. If the concepts are introduced in the later sections (like in (4)), you’d better considering rearranging them to improve your narrative.
8. Elaborate on your choice of $\Lambda$ in (3). What is your motivation? What does it indicate? Any related literature where this metric is used and explained in detail? What is its relation to the Lyapunov condition? You say “strong numerical evidence”, how strong? Provide introductions or references for your readers to understand.
9. Rephrase Line 304. Unclear. Better give a rigorous math expression for clarity.

### Section 4

1. The important variable σ is not introduced very clearly. The only related explanation is *“σ characterizing the root mean square of the off diagonal rows”*. I can hardly get what you mean. Please use a better expression.
2. You need to show your Persian rug’s optimality with experiments.
3. Does Eq. 8 imply we can fix b at any value and only adjust a to obtain the optimal algorithm? If not, how to select $b$?
4. Verify "This implies that even if ... scaling of the loss for $r>p$." on Line 490 with experiments.

## Minor

1. Always cite related works when you introduce new concepts, like Hadamard matrix, Lyapunov condition, etc.
2. Unnecessary verbosity. For instance:
    1. (Line 185 to 188) “…diagonals are large…as possible.” Why don’t you just say $W \approx c I$?
    2. (Line 22 to 25) “Unlike…oblivious”. Why should the algorithm care?

    You should flesh out your paper with rigorous and detailed definitions of your important concepts rather than these.

3. What do you mean by “r=0.25” in Fig. 2a? Is r defined on Line 424?
4. Line 178 “p = 4.5% and ratio ns = 512.” Is it a typo? Do you mean nd/ns = 512?
5. On line 204, please provide a clear definition of “mean-square fluctuation” or $\Delta$. Is it simply the variance (looks so on Line 212)? In addition, variance is not sufficient for evaluating if all elements in a set are nearly the same. You’d better additionally evaluate the maximum absolute difference. I also suggest replacing variance with standard deviation.
6. Why use $A$ rather than $a$ in Appendix A? Improve consistency.
8. What is $\mathbf{B}$ on Line 323 and 363? $\mathbf{b}$ in (1) and (2)? Improve consistency.
9. Use the correct citation command: \citep and \citet.
10. Use “Fig. xx” instead of “fig. xx” to refer to figures, and "Eq. xx" instead of "eq. xx" for equations.

**Questions:**

1. On line 111, why must we construct $x_i$ like this? Can we sample $u_i$ from the other distributions, like uniform[-1, 0]? This still makes $x$ sparse, right? I guess sampling $u_i$ from the other distributions may not work because you AE has ReLU at the output activation? Are you assuming that $x$ are outputs of ReLU or any other positive activation?
    * How well does this synthetic distribution of sparse data agree with the distribution of sparse data in reality? Here you are assuming its dimensions to be i.i.d, in reality this is probably not the case.
2. You said “We must restrict to W with rank no more than nd” on Line 324. Do we need to take this rank condition into account when deriving Equation (4) and making the following reasonings from Line 332 to 348? If not, why? Could $\nu_i$ correlate with each other due to the low rank of $W$? Are you making any i.i.d. assumption about the off-diagonal elements here?
    * Also, this “because νi becomes Gaussian” seems to be a conjecture based on your experiment in Section 3.2. I’m not familiar with your test so I don’t know how reliable it is. That said, you’d better make this assumption stand out with something like `\newtheorem{asp}{Assumption}`.
3. Given your Persian rug construction of $W$, any way to retrieve $W_\text{in}$ and $W_\text{out}$ from it? Are they unique? Can we use the pair on Line 398?
4. Expand your derivation of Eq. 19 in Appendix B to make it clearer. Why is there no $a$ in it (considering your definition of $\nu$ on Line 333)? Does this equation only hold when you construct $x_i$ as described on Line 111?

---

> ### Author Response · Authors · 2024-11-26
> **Official Response by Authors**
>
> We thank the referee for the review. Many of these points helped us clarify the writing.
>
> **Weaknesses:**
> The theoretical findings of Section 4 should be correct, especially given that our newly derived lower bounds are in agreement with the upper bounds. Nevertheless, we can also add plots that confirm the derived bounds. We will confirm here when this is complete. **Update: this is complete!**
>
> We have attempted to better explain the jargon and remove unnecessary verbosity. If there are any specific instances of this the reviewer would still like us to address, please let us know.
>
> The toy model of Elhage et al. 2022 is a well-studied model for understanding superposition. It is not a perfect model of LLMs because it only captures how features are linearly encoded in activations. While in practice language models also contain non-linearly encoded features, we believe the methods developed in our work on the linear case will generalize to future toy models that capture these features.
>
> # Major
>
> ## Abstract
> In our updated version of the paper we derive a lower bound on the loss valid for any elementwise or gating activation function in the appendix. This clarifies the point made in the abstract.
>
> ## Section 1
> By "permutation symmetric data" we meant that the distribution of $x\in\mathbb{R}^n$ is such that $(x_1,...,x_n)$ is equal in distribution to $(x_{\pi(1)},..., x_{\pi(n)})$ for any permutation $\pi$. By ``thermodynamic limit'' we mean the limit as $n_s\to\infty$. These terms are standard in some parts of the machine learning community, but we have now also defined them explicitly in our text.
>
>
> ## Section 2
> We trained using SGD online on randomly generated samples. More experimental details can be found in the appendix. (Once the authors are made public, we will also submit a version which has links to a public Github, for easy reproducibility.)
>
> ## Section 3
> 1. **Update: We have changed the wording. See also the appendix with empirical tests.**
> 2. Multi-line answer:
>
> Thank you for pointing this out, it should indeed be $n_s\to\infty$. We have fixed the typo.
> - We believe 8192 shows convergences to Gaussianity for most of the parameter space (see Figure 7 in the new paper or Figure 6 in the original one).
> - Yes, thanks for the pointer. We mention this fact in the definition of the model (the beginning of Section 2) now.
> 3. Thank you, we want the paper to be as precise and readable as possible. We have edited this portion of the paper (beginning of Section 3.2). (In this case, we say a vector is permutation invariant if $x_{i} = x_{\pi(i)}$ for any permutation $\pi$. Similarly, as above, we say a square matrix is permutation invariant $A_{ij}=A_{\pi(i)\pi(j)}$ for any permutation $\pi$.)
> 4. This sentence was indeed unclear and we removed it in the rewrite of this section.
> 5. As above.
> 6. We have made the references to plots more precise.
> 7. Multi-line answer:
>
> Indeed, we should have defined $\nu$ (or $\nu_i$ for the $i$'th row) earlier. We have done this now.
>
> Indeed the $\Delta$ notation may have been confusing since it's effectively a sample variance (e.g. you were correct to think $\Delta\text{var}(\nu)$ was the variance of the variances of $\nu_i$ over $i$). In the paper, we reserved the terms mean and variance for random variables, to help keep clear what is random and what is considered fixed in our theory. We rewrote this section and so it should be clearer now.
>
> 8. Multi-line answer:
>
> The quantity $\Lambda$ measures the distance of the error term $\eta_i$ to a Gaussian distribution. This being small justifies the use of the Gaussian approximation in Section 4.
>
> As the referee is likely aware, such Gaussian approximations are typically justified by central limit theorems (taking $n_s$ to infinity) or else some Barry-Esseen-type (finite $n_s$) bound. The quantity we give is actually such a finite $n_s$ bound and we give the relevant pointer to the theorem in the text. The quantity $\Lambda$ also arises in Lyapunov's condition for the central limit theorem; see Billingsley, "Probability and Measure" or Durrett, "Probability: Theory and Examples" (the latter is open access and easily accessible online, but the theorem is left as an exercise in that book).
>
> 9. Done!
>
> ## Section 4
> 1. The new version makes clear that $\sigma^2$ is the variance of $\nu$.
> 2. We show how the Persian Rug construction achieves a newly derived lower bound on the loss in the appendix now, proving optimality.
> 3. Equation 8 does not imply that we can fix any value of $b$, but rather that for a given value of $b$ we can find the optimal value of $a$. In general fitting the optimal value of $b$ must be done numerically, either by optimizing the expected or empirical loss.
> 4. In our updated draft we show a general lower-bound which, in combination with the upper bound, shows that the error is controlled by the level of noise, $\sigma$.

---

> ### Author Response · Authors · 2024-11-26
> **Official Response by Authors (Part 2)**
>
> # Minor
> Mostly done, but will confirm these later.
> **Update:** Thanks for finding these issues. We made most of the proposed changes, except for a few minor stylistic ones. We also checked the maximum absolute difference and found it was converging to zero as well (see appendix).
>
> # Questions
> 1. In this context, assuming sparse data of this form is standard (see Elhage et al. 2022, Ganguli and Sompolinksy 2010, and also the references in reviewer ru6M's comments on our work). Changing to $[-1, 0]$ won't matter in a practical case as this corresponds to flipping the sign of the embedding vectors.
>
>  - Our work demonstrates that models become easier to analytically analyze when their weights inherit symmetries in their training data. While we considered the maximal case of permutation symmetry, a similar simplification of the loss occurs if the weights inherit smaller symmetries. As a demonstration we have added such an example to the appendix. Real world data is of course significantly more complicated, but we nevertheless expect model weights to reflect its structure. However, we believe a first principles understanding of how this occurs requires first the development of fundamental tools to analyze toy models of increasing sophistication. See also [this](https://www.alignmentforum.org/posts/uvEyizLAGykH8LwMx/fundamental-vs-applied-mechanistic-interpretability-research) discussion on the differences between fundamental and applied mechanistic interpretability.
>
> 2. The rank condition need not be considered when deriving equation (4) since it is just an identity which arises from the definition of $\nu_i$. The $\nu_i$ certainly correlate with each other, but because each output is purely a function of $x_i + \nu_i$ this fact doesn't matter. No assumptions are made about the off diagonals.
>
> - The current version of the paper should be more clear on this point now: the empirical observations of Section 3 are used as assumptions in Section 4. See the discussion regarding the Barry-Esseen theorem above as well.
>
> 3. One way to recover $W_{\text{in}}$ and $W_{\text{out}}$ from $W$ (generally) is to perform an SVD so that $W=UDV$. $W_{\text{out}}$ can then be written as $U\sqrt{D}$ where $\sqrt{D}$ is an $n_s \times n_d$ matrix with diagonals equal to the (positive) square-roots of the non-zero singular values of $W$. Similarly $W_{\text{in}} = \sqrt{D}V$. This is not unique because $W = W_{\text{out}} W_{\text{in}} =  (W_{\text{out}}M)(M^{-1} W_{\text{in}})$ for any invertible $M$. This shows that we could equally well have taken different $W_{\text{in}}$ and $W_{\text{out}}$. The pair on line 398 would indeed work.
>
> 4. For this section we restricted to $W$ which have 1 on the diagonal rather than $a$ for ease of notation. It's correct that this equation is particular to our setting, in particular the facts that the variance of $x_i$ is $(4p - 3p^2)/12$ and that the $x_i$ are i.i.d.

---

> > ### Comment · Reviewer_iN6D · 2024-11-27
> >
> > I appreciate the authors' effort in addressing my concerns. I believe this revision is now clear enough to bring some intellectual contributions to the field. Thus I adjust my score accordingly.

---

### Official Review · Reviewer_f6ZT · 2024-11-01

**Soundness:** 4
**Presentation:** 4
**Contribution:** 2
**Rating:** 6
**Confidence:** 3

**Summary:**

The authors study the compression of sparse data using a two-layer autoencoder with ReLU output non-linearity, considering the toy model of Elhage et al (2022). Building on an ansatz for the product of the weights, supported by numerical observation, they derive a closed-form expression for the population loss depending on only three scalar parameters, corresponding to the bias, common diagonal element of the product matrix, and variance of off-diagonal elements. They deduce from it a lower-bound on the loss, and construct a deterministic weight matrix that achieves it. They find that the corresponding loss coincides with that of the trained model.

**Strengths:**

On a technical level, the authors provide a detailed study of the optimal population loss achieved by the considered model, which is non trivial given the ReLU non-linearity. To the best of my awareness, this derivation is new compared to the original work of Elhage et al. (2022). The finding of the weight matrix achieving the same loss as the model optimized using classical gradient-based optimizers is an interesting construction. Although I have not checked the math in detail, the derivation looks scientifically sound.

On a writing level, the paper is very clearly written, with sufficient discussion and intuition building. The assumptions are illustrated by careful numerical experiments.  Sufficient discussion accompanies all important equations.

**Weaknesses:**

My main concern lies in the scope of the contribution. Some of my concerns are tied to my questions, which I list in the next section.

- To my understanding, the study is confined to the isotropic case of Elhage et al. (2022), as it builds upon the ansatz that all diagonal elements of $W_{out}W_{in}$ are equal. This means there is not really any feature learning in the model, although the considered setting remains non-trivial.

- While the study is interesting, it does not add much more phenomenological insights compared to the original work. In particular, the authors construct a matrix which achieves the same optimal population loss as the trained model, which is one of the strengths of the work. However, I understand any matrix of the form (7) is similarly optimal, and I do not understand why the explicit realization in 4.2.2 brings any further insight. I might have misunderstood a point, and am happy if the authors bring clarifications on this point.

Overall, I think the study is technically sound, the insights tied to the analysis of interest, but brings little novel phenomenological insight.  I am thus giving an accept score -- because the paper looks sound --, but not a high one.

**Questions:**

- (Minor) Do the authors believe the study could be extended to non-isotropic cases, where some features are of higher importance, e.g. in the block case where blocks of different features have the same importance ?

- Is it correct that (7) also corresponds to the weights a linear model would learn ? If that is the case, does the improvement in reconstruction loss from the reLU model come from the bias and activation alone ?

- As pointed above, is there any specific reason the particular realization of 4.2.2 of optimal weights is of specific interest, compared to randomly drawing an orthogonal matrix $O$ and constructing a weight matrix as in (7)? Do the authors claim the model learns weights exactly corresponding to 4.2.2, or simply that the loss coincide ?

---

> ### Author Response · Authors · 2024-11-24
> **Official Comment by Authors**
>
> We thank the reviewer for their careful review and answer their questions and address the weaknesses brought up.
>
>
> **Weaknesses:** While the original work by Elhage et al. (2022) empirically reveals a number of important phenomena appearing in the toy model of superposition, the work does not explore the network's exploitation of randomness, which is reflected in their focus on smaller models. More explicitly, our experiments show permutation-symmetry emerges only in sufficiently large values of $n_s$. The utility of randomness in large systems is familiar to physicists, and it is our hope to convey this to the mechanistic interpretability community, who we feel often overlook this effect by focusing purely on small-scale, detailed structures. Thus while it is true that any matrix of the form in (original submission 7) is similarly optimal (so long as it leads to equal variances for all $\nu_i$), the purpose of the explicit construction is to demonstrate that such matrices do exist, and to highlight that the detailed structure of a matrix may not be important when doing interpretability.
>
> **Question 1:** Yes, we now have empirical evidence that our study can be extended to non-isotropic cases in two ways: smaller symmetry groups and perturbations around a fixed symmetry.  We empirically observe that model weights inherit statistical symmetries even for smaller symmetry groups, such as permutations within individual blocks. This implies a similar simplification to the loss. The second way concerns small perturbations around a given symmetry, for which we find the solution is stable. We have added a discussion of this to the the appendix in the updated manuscript.
>
> **Question 2:** A linear model will not learn the same set of weights. Most importantly, a linear model will learn any (unscaled) rank $n_d$ projection, while the non-linear model will learn a scaled projection with equal diagonals. More generally, if some features are more important than others, the linear model's projection will be along the largest eigen-directions of the sample covariance matrix, with the bias $\mathbf{B}$, fitting the mean of the data. The way to interpret this is that a linear model cannot store more directions than $n_d$ while adding a ReLU allows the model to leverage the sparsity of the input and store more information about the input.
>
> **Question 3:** Randomly drawing an orthogonal matrix will not allow us to satisfy the condition $W_{ii}$ are constant. A random projection works reasonably well, but the diagonal elements still fluctuate, and this leads to a greater error than necessary. Intuitively this is because the diagonal elements control the strength of the signal term $x_i$ which is particularly important. The purpose of the particular construction in 4.2.2 is to demonstrate that such matrices do exist, and to highlight the algorithm's insensitivity to small-scale structure in the $W$ matrix and in particular its exploitation of randomness.

---

> > ### Comment · Reviewer_f6ZT · 2024-11-25
> > **Acknowledgement of rebuttal**
> >
> > I wish to thank the authors for their detailed reply and clarifications. I stand by my original evaluation that the paper seems technically sound (although I have not checked the derivation in detail), but I again share the other reviewer's concerns about the scope of the paper and its interest to the broader ICLR community.

---

### Official Review · Reviewer_br6G · 2024-11-03

**Soundness:** 3
**Presentation:** 3
**Contribution:** 3
**Rating:** 6
**Confidence:** 3

**Summary:**

The paper both theoretically and empirically performed an exhaustive analysis of autoencoder trained on sparse data,
extracting the essential structure needed to represent such data. The paper firstly reveals that the change of loss as the compression ratio varies and shows that when compression ratio is near zero, the loss drops to zero entirely. This motivates the authors to consider the structure of the weight in both linear and non-linear cases. They artificially include the statistical permutation symmetry into the weight and found the artificial weight and the trained model returned the same result. The conclusion is they qualitatively show that a statistically symmetric strategy exists in certain regimes of the macroscopic parameters. Thus optimal choice for W exisits based on the proposed  Persian Rug Model.

**Strengths:**

S1. Novelty: The paper is novel in it both theoretically and empirically performed exhaustive analysis of autoencoder trained on sparse data, interpreting the essential structure needed to represent such data. I like the way they demonstrates the distribution of the weights.

S2. The paper firstly reveals that a statistically symmetric strategy exists in certain regimes of the macroscopic parameters and thus optimal choice for W exisits based on the proposed  Persian Rug Model. The theoretical support of the motivation is strong.

S3. The presentation of the paper is good, with a good clarity.

**Weaknesses:**

W1. What is the interpretation of the model if the input is not sparse? Is the symmetry property and the relevant conclusion partially hold? I noticed the theories are mostly based on the compression ratio. Does this mean the interpretation will change if the compression ration becomes high?

W2. What is the significance of the model if the input is no longer sparse?

**Questions:**

My questions are mostly overlapped with the weakness section. I appreciate it if the authors may further instantiate the following points:

W1. What is the interpretation of the model if the input is not sparse? Is the symmetry property and the relevant conclusion partially hold? Does this mean the interpretation of the weights and the optimal W will change significantly if the compression ration becomes high?

W2. Generally speaking, what is the significance of the proposed model if the input is no longer sparse, whereas in most cases the input is not sparse?

---

> ### Author Response · Authors · 2024-11-24
> **Official Comment by Authors**
>
> We thank the reviewer for their time and their review. Here are the answers to the questions brought up.
>
> **Q1:** When the input data is not sparse then it cannot be compressed into a small subspace. Mathematically when $p n_s > n_d$ then the number of sparse features which are on is typically greater than the number of dimensions available to store them. This is explained via our theory because the variance of $\nu$ is large and therefore $x + \nu$ has little information about $x$. More precisely, this means that the conditional distribution $P[x | x + \nu]$ becomes broad and is therefore not informative about the true value of $x$. Therefore the most interesting regime is the one where $p n_s < n_d$, where compression is possible (though still non-trivial). The symmetry property is preserved even in the non-sparse case, however we do not focus on this regime.
>
> **Q2:** Our interest in this model is specifically related to understanding the behavior of sparse autoencoders, which take a distribution over vectors $x \in \mathbb{R}^{n_d}$ (which is not sparse) and find an equivalent representation $y(x) \in \mathbb{R}^{n_s}$ where $y(x)$ is sparse and there is some linear map $A$ such that $x = Ay(x)$. We consider the problem where the ground truth of the $y$ distribution is known, and as such that distribution is naturally sparse.

---

> > ### Comment · Reviewer_br6G · 2024-11-25
> > **Thanks for the response**
> >
> > Thanks for the response to my questions. As the authors have addressed my questions, I will maintain my score as a weak acceptance. I noticed my colleague has some theoretical concern about the paper, but generally, I think this paper has certain contribution to the community.

---

### Official Review · Reviewer_ru6M · 2024-11-06

**Soundness:** 2
**Presentation:** 2
**Contribution:** 2
**Rating:** 3
**Confidence:** 4

**Summary:**

This paper characterizes the solutions of training sparse autoencoders (defined as $f(\mathbf{x}) = \textrm{ReLU}(W_{out} W_{in} \mathbf{x} + \mathbf{b})$) with the reconstruction loss $L=\mathbb{E} || \mathbf{x} - f(\mathbf{x}) ||^2$. The paper first trains an autoencoder on synthetic data consisting of sparse vectors generated from a Bernoulli-Uniform model and records some empirical observations about the composite matrix $W=W_{out} W_{in}$. The authors note that the off diagonal entries of this matrix seem to follow a normal distribution that seems to be identical across rows and that the diagonal entries are also identical. This allows them to reduce the loss function to a function of three scalar variables corresponding to the diagonal, off-diagonal, and bias terms. The solution they obtain is of the form $W = OPO^\top$ where $O$ is orthogonal and $P$ is a rank $n_d$ projection matrix. The authors also characterize how the loss varies as a function of the Bernoulli sparsity parameter $p$.

**Strengths:**

The empirical observations and theoretical results are reasonable and the authors provide an adequate description of the solution $W$.

**Weaknesses:**

Connections between sparse coding and autoencoders have been explored extensively in the literature. This paper does not cite or engage with any of this literature.

The empirical observations about $W$ having identical diagonal entries and the off diagonal entries being small and having a common distribution can be captured by $W = A^\top A$ where $A \in \mathbb{R}^{n_d \times n_s}$ is an overcomplete, incoherent dictionary with unit norm columns. This ensures that the diagonal terms are identical while the off-diagonal terms are small in magnitude. Moreover it is well known that random matrices satisfy the incoherence/restricted isometry condition and a random $A$ ensures that the off-diagonal entries follow the same distribution.

This model of dictionary learning, along with Bernoulli-Uniform or Bernoulli-Gaussian sparse codes has been studied in the complete and overcomplete setting with neural and non-neural algorithms [3,4,5].

Reference [1] studies how ReLU autoencoders with tied weights can learn dictionaries with data from this generative model. The authors show (in theorem 3.1) that one layer of a ReLU autoencoder can recover the support of the sparse code with high probability if the weights are close to the ground truth dictionary. The result also proposes a bias vector with all negative entries whose magnitude can be derived from the incoherence of the dictionary. The paper also uses a landscape argument to show that the ground truth dictionary is a critical point for the reconstruction loss. Reference [2] goes further to show that autoencoders trained to minimize reconstruction error with gradient descent can recover the ground truth dictionary.

[1] Rangamani, A., Mukherjee, A., Basu, A., Arora, A., Ganapathi, T., Chin, S., & Tran, T. D. (2018, June). Sparse coding and autoencoders. In 2018 IEEE International Symposium on Information Theory (ISIT) (pp. 36-40). IEEE.

[2] Nguyen, T. V., Wong, R. K., & Hegde, C. (2019, April). On the dynamics of gradient descent for autoencoders. In The 22nd International Conference on Artificial Intelligence and Statistics (pp. 2858-2867). PMLR.

[3] Arora, S., Ge, R., Ma, T., & Moitra, A. (2015, June). Simple, efficient, and neural algorithms for sparse coding. In Conference on learning theory (pp. 113-149). PMLR.

[4] Agarwal, A., Anandkumar, A., Jain, P., & Netrapalli, P. (2016). Learning sparsely used overcomplete dictionaries via alternating minimization. SIAM Journal on Optimization, 26(4), 2775-2799.

[5] Spielman, D. A., Wang, H., & Wright, J. (2012, June). Exact recovery of sparsely-used dictionaries. In Conference on Learning Theory (pp. 37-1). JMLR Workshop and Conference Proceedings.

[6] Refinetti, M., & Goldt, S. (2022, June). The dynamics of representation learning in shallow, non-linear autoencoders. In International Conference on Machine Learning (pp. 18499-18519). PMLR.

While these prior results study a slightly different model architecture, they can be adapted to the model studied in this paper with a little effort. In my opinion the results obtained by the current paper can be explained by the sparse coding model, and it is not true that this has not been studied in the context of autoencoders.

**Questions:**

1. How does the solution proposed by the authors differ from the dictionary learning solution? Can the authors delineate conditions under which trained autoencoders converge to their solution instead of overcomplete, incoherent dictionaries?

2. The previous literature does not study autoencoders and sparse coding in the context of interpretability. Are there specific questions that arise in this context that require the authors' model? Does the authors' analysis answer these questions?

---

> ### Author Response · Authors · 2024-11-24
> **Official Comment by Authors**
>
> We thank the reviewer for an in-depth and constructive analysis of our work, and for pointing us to related literature on sparse-coding and autoencoders. We believe it is indeed worth engaging with this work, and have added their relationship to our related works section. However, in contrast to the reviewer's suggestion, we believe this work can not be trivially adapted to arrive at our conclusions, and furthermore that the techniques by which we solve the architecture are useful for a wider class of problems in mechanistic interpretability. We justify this claim in the following answers to the reviewer's questions.
>
> The references [1-5] have to do with the sparse coding problem, which (as the reviewer is aware) is very related to but not the same as the toy model of superposition we study in this paper. For easy reference, we re-state these problems now.
>
> Problem 1 (Sparse Coding; from [3]): Given a set of vectors $y^{(1)}, y^{(2)}, ..., y^{(p)} \in\mathbb R^{n}$, find a sparse coding matrix $A\in\mathbb R^{n\times m}$ and sparse coefficient vectors $x^{(1)},...,x^{(p)}\in\mathbb R^{m}$ that minimize the the following reconstruction error with sparsity penalty
>
> $$
> \mathcal E (A,X) = \sum_{i=1}^p \left(||y^{(i)} - Ax^{(i)}||_2^2 + S(x^{(i)})\right).
> $$
>
> Problem 2 (Toy Model of Superposition): Given a set of sparse vectors $x^{(1)},.., x^{(p)}\in\mathbb R^m$, find a sparse coding matrix $A\in\mathbb R^{n\times m}$ and decoder matrix $B\in\mathbb R^{m\times n}$ that minimizes the reconstruction error
>
> $$
> \mathcal{F}(A,B,X) = \sum_{i=1}^p ||\text{ReLU}(BAx) - x||_2^2.
> $$
>
> (Note that we write a slightly different variant of the toy model problem compared to the one in our paper for easy comparison with the Problem 1.)
>
> The papers [1-5] assume the $y^{(i)}$ are generated by some true dictionary $A^*$ and coefficients $x^{(i)}$ that satisfy certain properties (e.g. that $A^*$ is incoherent) and then show that these assumptions imply various desirable properties for various algorithms for learning $A$ and $x^{(i)}$ (for example, convergence of the optimizer to the true generating dictionary $A^*$).
>
>
> In contrast, the neural networks in our work search over the space of dictionaries to find ones that encode sparse information in a way particularly suitable for reconstruction by a single linear + ReLU layer. As a result, the dictionary our network finds contains \textit{additional structure optimized for a recovery process} consisting of a single linear layer followed by local activation function. In particular we find that the relevant error parameter is not the incoherence $(\max_{i\neq j} |W_{ij}|)$ but rather the variance of off diagonal elements in each row $( \sigma=\sum_j W_{ij}^2)$, and that it is this parameter that needs to be minimized for a given compression ratio, and this leads to the additional non-trivial structure such as $W$ having being a projector with equal diagonals. This structure is not present in standard constructions of incoherent dictionaries. Choosing unit vectors randomly will fail to suppress $\sigma$ when $n_d\sim O(n_s)$,  while a random projector will fail to suppress diagonal fluctuations sufficiently quickly as $n_d/n_s\rightarrow 0$.
>
> The referee has proposed that the methods that are developed in the papers [1-5] for problem 1 should easily transfer to problem 2, even if the problems are somewhat different. We investigated this question and believe this is not the case for the following reasons.
>
> The key difficulty in adapting these methods to our case is that, in our case, the corresponding matrix to $A^*$ (namely $W_{in}$) is changing over the learning process, and it is not clear the assumptions [1-5] put on $A^*$ will be satisfied. (For example, [3] assumes that $A^*$ is incoherent and $||A^* ||$ is suitably bounded, [4] requires a $A^*$ to satisfy a bound on its RIP constant and has bounded norm, and [5] requires $A^*$ to be square (which is definitely not true in our setting)). Establishing that these conditions hold over the course of an optimization process seems like interesting future work, but is not the point of this paper. It is also not immediately apparent that the theoretical machinery developed to leverage these assumptions will apply when the dictionary $A^*$ is changing, even if the assumptions hold over the course of training (or eventually).

---

> > ### Comment · Reviewer_ru6M · 2024-11-26
> > **Concerns remain**
> >
> > 1. Reference [1] in my original review studies the minimization of the objective $L(W) = \frac{1}{2} \sum_{i=1}^p || y^{(i)} - W^T \textrm{ReLU}(Wy^{(i)} - \epsilon) ||^2$. It is this objective I believe is quite closely related to the toy model of superposition objective.
> >
> > 2. "As a result, the dictionary our network finds contains _additional structure optimized for a recovery process_ consisting of a single linear layer followed by local activation function." The additional structure claimed here by the authors is a projector with equal diagonals, and small off-diagonal elements. This structure can also be obtained by $W = A^{\*\top}A^\*$ where $A^*$ is an incoherent dictionary. If we consider an $A^*$ that has unit norm columns, the diagonal elements of $W$ will all be equal to 1. The off diagonal elements of $W$ will all be bounded by $\frac{\mu}{\sqrt{n}}$. Does this not satisfy the requirements for reconstructing the sparse code with one layer of an autoencoder?
> >
> > 3. The current paper also conducts a landscape analysis of the optimal solution, and does not analyze the entire optimization trajectory. The claim that insights from sparse coding are not applicable here is not convincing since these insights can also be applied in the case of a landscape analysis (as done in [1]). If the authors would like to argue this point, I would point out that they do not provide evidence for $W$ being close to a projection matrix throughout the optimization trajectory. Moreover the incoherent properties are not required to hold across the entire optimization trajectory. Reference [2] shows convergence to an incoherent dictionary without this assumption.
> >
> > I will keep my current score.

---

> ### Author Response · Authors · 2024-11-26
> **Addressing point 2**
>
> We here address the most crucial concern, [2]. The point we wish to make is that there is additional structure beyond the fact that $A^*$ is an incoherent dictionary. While it is true that choosing $A^*$ to have unit norm columns will give diagonal elements of $1$, it will not result in a $W$ which is a projector. The latter condition gives a pattern of off diagonals which optimally interfere when acting on the input vector.
>
> To make this point clearer, let us consider two incoherent dictionaries $A^1$ and $A^2$ such that the corresponding $W^1$ and $W^2$ both satisfy ($|W^1_{ij}| = |W^2_{ij}| = \epsilon$) for all $i\neq j$. However, for $W^2$, in each row half of the off diagonal elements are positive and the other negative, while for $W^1$ all elements are positive. Consider now the action of the first row of each $W$, $W^1_{1 \bullet}$, and $W^2_{1 \bullet}$, onto the all ones vector $\vec{1}$. In the first case we get $W^1_{1 \bullet}\cdot\vec{1} = 1+\epsilon (n_s-1)$, while in the latter case we obtain  $W^2_{1 \bullet}\cdot\vec{1} = 1$. We see that in the latter case the oscillation of signs allows for the total cancellation of off diagonal elements, and that knowledge of the coherence parameter was not sufficient to determine how distorted the signal would be.
>
> In summary, our model does not simply make use of the off diagonals having small absolute value as would be true for a typical incoherent dictionary, but also optimizes the distribution of signs in order to maximally suppress the noise. Since we are interested in recovering continuous signals as accurately as possible, this makes a large difference in certain parameter regimes.

---

> > ### Comment · Reviewer_ru6M · 2024-11-26
> >
> > I see two issues with your example:
> >
> > 1. In your example the matrix $W^1$ has smaller variance of the off-diagonal elements than $W^2$, but it sounds like $W^2$ is your preferred matrix (contra the previous response)? Do we prefer smaller or larger variance?
> >
> > 2. The more serious issue is that the vector of ones is not a sparse vector, and we should not expect a sparse autoencoder to reconstruct it perfectly. For sparse vectors the off diagonal elements should sum to $pn_s \times \frac{\mu}{\sqrt{n_d}}$, not $n_s \times \frac{\mu}{\sqrt{n_d}}$. As long as the vectors are sparse enough ($pn_s < \sqrt{n_d}$), this should not be as large or larger than the magnitude of the sparse code ($\Theta(1)$). In other words, we do not require the matrix $W$ to be a full isometry. We only need it to be a _restricted isometry_ over the set of sparse vectors. This is a property satisfied by incoherent matrices (and random matrices). So the question remains - why is your solution the only preferred solution?

---

> ### Author Response · Authors · 2024-11-26
> **Clarifying the purpose of the example**
>
> The example was meant to illustrate the conceptual point that one must specify additional information beyond the fact that $A^*$  is an incoherent dictionary in order to obtain the optimal solution. Addressing the critical concern:
>
> 2. That our solution is the preferred symmetric solution for this architecture is proved in E.2 where we show a lower bound on the loss for architectures of this form together with 4.2.1 and appendix D where we derive the form of the optimal matrix. Because the loss is lower bounded above zero the reconstruction is necessarily imperfect, and there is no exact sense in which any particular construction does or does not satisfy a restricted isometry property, but only the degree to which it minimize the loss. The form we derive is provably optimal among symmetric solutions, which are the preferred solutions of these neural networks. We can illustrate the difference between our construction and the reviewers proposed construction (random matrices) by considering the extreme case of $n_d = n_s$. Our construction gives $W = I$, while the reviewers construction will give a $W$ with small off diagonal terms. The performance of these two solutions will converge in the limit $p\to 0$ and differ drastically near $p = 1$. Our solution correctly characterizes the network's learned behavior across the entire parameter space of $p\in \[0,1\]$. Note that the difference in performance between the two constructions remains as we move slightly away from the extremal limit, such as  $n_d = n_s - k$ for $k \ll n_s$, where the explicit form of the optimal $W$ is no longer obvious.

---

### Meta-Review · Area_Chair_uoSR · 2024-12-15

**Metareview:**

The paper studies the compression of sparse data via a two-layer autoencoder with ReLU output non-linearity. Building on an ansatz for the product of the weights (supported by numerical evidence), the authors derive a closed-form expression for the population loss depending on three scalar parameters. This leads to a lower-bound on the loss, which is achieved by a certain deterministic weight matrix.

Reviewers ru6M and dnUr have raised important issues concerning the absence of comparisons with relevant related work (providing a rather extensive list of references) and the technical contributions themselves (no analysis of global optimality; little justification about the parameterization considered in the paper; unclear whether the proposed solution actually occurs in practice). Such concerns were unfortunately not resolved during the discussion phase and therefore I recommend a rejection at this stage.

I do encourage the authors to address the comments of the reviewers in a thorough revision and submit an improved version to a future venue.

**Additional Comments On Reviewer Discussion:**

Reviewer ru6M raised important concerns about the technical contributions and about the absence of discussion/comparison with related work. Reviewer dnUr also raised concerns along similar lines. These issues were not resolved during the rebuttal and they lead me to a 'reject' decision.

---

### Decision · Program_Chairs · 2025-01-22

Reject